behaviour, ecology

Allee effect, bioacoustics, conservation biology, captive breeding, animal behaviour

**Author for correspondence:**
Ross Crates
e-mail: ross.crates@anu.edu.au

# Loss of vocal culture and fitness costs in a critically endangered songbird

Ross Crates[1], Naomi Langmore[2], Louis Ranjard[2], Dejan Stojanovic[1], Laura Rayner[1], Dean Ingwersen[3] and Robert Heinsohn[1]

[1]Fenner School of Environment and Society, Australian National University, Linnaeus Way, Acton, Canberra 2601, Australia
[2]Research School of Biology, Australian National University, 46 Sullivan's Creek Rd, Acton, Canberra 2601, Australia
[3]BirdLife Australia, Carlton, Victoria 3053, Australia

  RC, 0000-0002-7660-309X; NL, 0000-0003-3368-6697; DS, 0000-0002-1176-3244; RH, 0000-0002-2514-9448

Cultures in humans and other species are maintained through interactions among conspecifics. Declines in population density could be exacerbated by culture loss, thereby linking culture to conservation. We combined historical recordings, citizen science and breeding data to assess the impact of severe population decline on song culture, song complexity and individual fitness in critically endangered regent honeyeaters (*Anthochaera phrygia*). Song production in the remaining wild males varied dramatically, with 27% singing songs that differed from the regional cultural norm. Twelve per cent of males, occurring in areas of particularly low population density, completely failed to sing any species-specific songs and instead sang other species' songs. Atypical song production was associated with reduced individual fitness, as males singing atypical songs were less likely to pair or nest than males that sang the regional cultural norm. Songs of captive-bred birds differed from those of all wild birds. The complexity of regent honeyeater songs has also declined over recent decades. We therefore provide rare evidence that a severe decline in population density is associated with the loss of vocal culture in a wild animal, with concomitant fitness costs for remaining individuals. The loss of culture may be a precursor to extinction in declining populations that learn selected behaviours from conspecifics, and therefore provides a useful conservation indicator.

## 1. Introduction

Cultures comprise evolved traditions that are maintained through information sharing between associates [1]. Culture plays an important role in identifying appropriate mates, predator–prey dynamics, innovation spread and dispersal [2,3]. Language in humans and songs in birds and mammals are vocal cultures learned from tutors in early life [4–6], making early life interactions critical for the maintenance of vocal cultures that enable effective communication between associates [4]. If vocal development is compromised by infrequent interactions with appropriate tutors, it could affect the maintenance of population-level vocal culture [7,8]. Evidence that vocal cultures can erode in small or sparse populations is well documented in humans [9], but limited in other species, despite many animal populations increasingly occurring at lower density [10–14]. Whether the loss of vocal culture can incur fitness costs and exacerbate population decline remains poorly understood.

Songbirds learn their songs from tutors in the natal area or shortly after dispersal [15,16], which usually become fixed after 1 year of life [17]. Songs facilitate mate attraction and territorial defence, so the accuracy of song learning can translate to substantial fitness outcomes [4,18,19]. If maladaptive song learning due to

Proc. R. Soc. B 288: 20210225

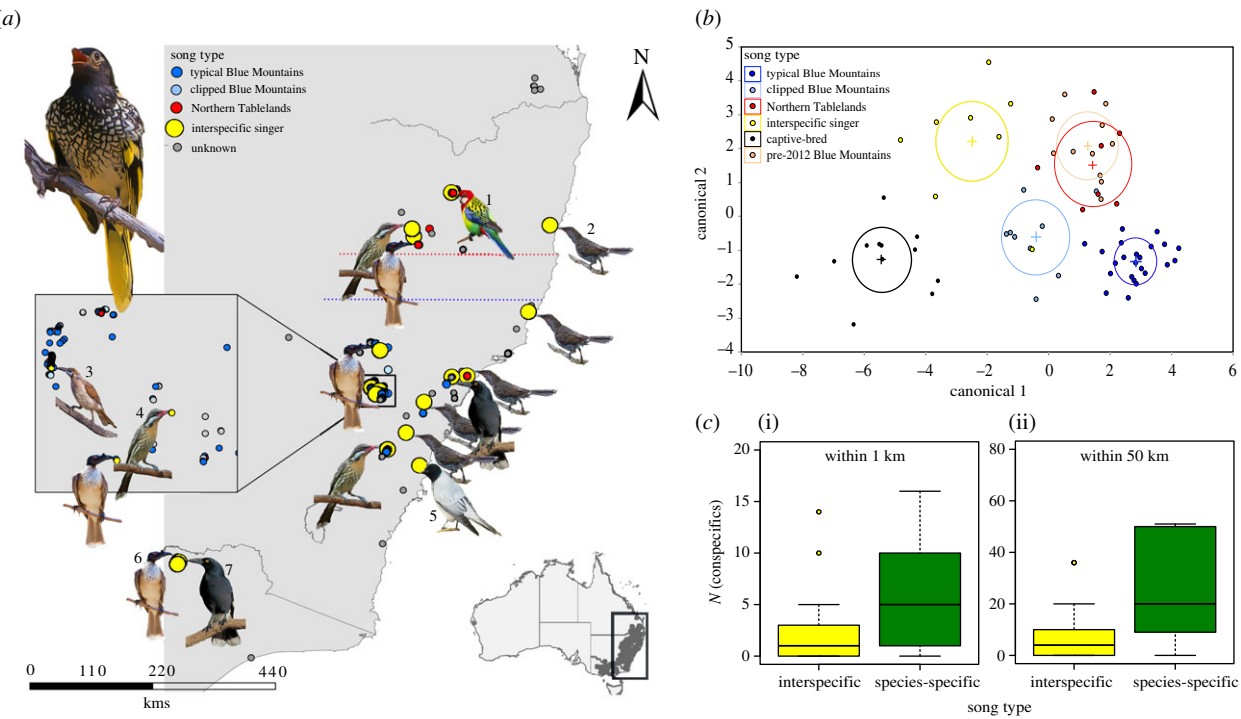

**Figure 1.** Spatial and acoustic summary of regent honeyeater song types. (*a*) Locations of contemporary wild male regent honeyeaters (2015–2019) and their song types. The species whose songs each interspecific singing regent honeyeater most closely resembled are shown: (1) eastern rosella; (2) little wattlebird; (3) little friarbird; (4) spiny-cheeked honeyeater; (5) black-faced cuckooshrike; (6) noisy friarbird; (7) pied currawong (electronic supplementary material, text S5). Dotted lines denote the southern and northern limits of distinct breeding areas in the Northern Tablelands (red) and the Blue Mountains (blue), respectively. Centre left inset: data from Capertee Valley, the core breeding area within the Blue Mountains. Bottom right inset: Location of study area on a national scale, with the regent honeyeater's contemporary range shaded dark. Due to map scale and spatial clustering of sightings, not all individuals are visible. (*b*) Discriminant function analysis of regent honeyeater song types, including captive-bred and pre-2012 birds from the Blue Mountains. Discriminant function analysis labels each multivariate mean with a circle corresponding to a 95% confidence limit for the mean. Groups that are significantly different have nonintersecting circles. (*c*) The number of contemporary wild, co-occurring male regent honeyeaters detected within the same breeding season within (i) 1 km and (ii) 50 km for male regent honeyeaters with interspecific songs (yellow) versus males with a species-specific song type (green). (Online version in colour.)

lower population density incurs fitness costs, it could decrease population growth rates as an Allee effect [20]. The process of song learning also has implications for reintroduction efforts; failure to expose captive-bred juveniles to appropriate song tutors could result in abnormal song production [21], potentially compromising the ability of captive birds to pair with wild conspecifics and reproduce post-release [22].

We explored the impact of severe population decline on song culture in a nomadic, nectarivorous songbird and the subsequent fitness consequences for remaining individuals. Until the mid-twentieth century, regent honeyeaters moved long distances (hundreds of kilometres) throughout south-eastern Australia in flocks of hundreds [23]. Widespread habitat loss has seen the population decline to an estimated 200–400 birds, distributed sparsely and dynamically through their 300 000 km² range [24]. Only male regent honeyeaters sing a full song, which they use during courtship and territory establishment [25]. Sons do not learn songs from their fathers, because adult males do not sing while they rear their offspring and chase independent juveniles from the natal area before they commence song production [17]. As for many songbird species [26], young male regent honeyeaters need to associate with older males other than their fathers in order to learn songs appropriate for their species and region.

By combining historical recordings with citizen science data and 5 years of standardized population monitoring, we quantified spatio-temporal differences in regent honeyeater song and song complexity within and between wild and captive-bred birds. We then examined the relationship between song type and wild population density, before assessing the fitness consequences of male song type in the wild population.

## 2. Methods

### (a) Study species

The effective regent honeyeater population is a single genetic unit of approximately 100 pairs [27,28]. Although birds are sighted occasionally throughout their range, most contemporary breeding is restricted to the greater Blue Mountains—estimated population 150 to 300 individuals, and the Northern Tablelands—estimated population 50 to 100 individuals, in New South Wales [25] (figure 1*a*).

When nesting, regent honeyeaters compete with conspecifics and other honeyeater species for small breeding territories, historically in loose aggregations [29]. Paired male regent honeyeaters associate closely with their partner female, cease singing following egg laying, and do not sing again until juveniles are independent 2–3 weeks post-fledging. Unpaired males do not associate closely with a female, tend to sing more frequently than paired males and are often chased away by any paired males present nearby [25]. Nest success is simple to quantify once the first egg or hatch date is known, as failed breeders typically disperse from the nesting territory shortly after nest failure, before the predicted fledge date. Successful breeders remain close to the nest for the fortnight post-fledging. The presence of juveniles can be determined via their begging calls or the alarm calls of their parents [25]. Birds breed from 1 year of age and mean lifespan is 6 years [24].

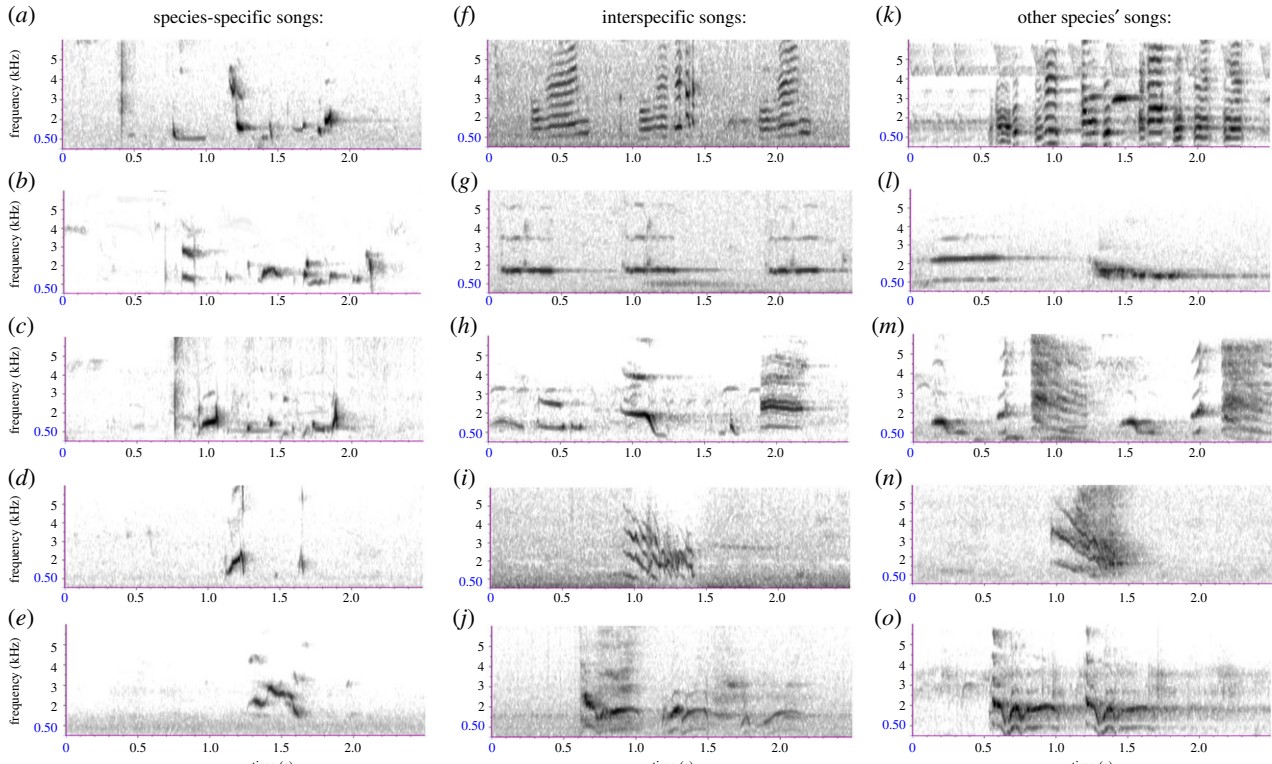

**Figure 2.** Spectrograms of regent honeyeater song types and the songs of the other species that the songs of interspecific singing regent honeyeaters most closely resembled. (*a–e*) Species-specific regent honeyeater songs: (*a*) pre-2012; (*b*) Northern Tablelands; (*c*) typical Blue Mountains; (*d*) clipped Blue Mountains; and (*e*) captive-bred. (*f–j*) Interspecific singing regent honeyeater songs: (*f*) noisy friarbird; (*g*) spiny-cheeked honeyeater; (*h*) little friarbird; (*i*) black-faced cuckooshrike and (*j*) pied currawong. (*k–o*) Other species' songs, which the interspecific singing regent honeyeaters closely resembled: (*k*) noisy friarbird; (*l*) spiny-cheeked honeyeater; (*m*) little friarbird; (*n*) black-faced cuckooshrike and (*o*) pied currawong. See electronic supplementary material, text S6 and S7 for further information on other species' songs and spectrograms, respectively. (Online version in colour.)

## (b) Data collection

We used data from all regent honeyeater sightings throughout the species's range from July 2015 to December 2019 to estimate the distribution and density of the remaining wild population. Regent honeyeaters can be sexed in the field based on a combination of their size, plumage traits, behaviour, vocal attributes and in the hand (via differences in wing length and body mass) during marking with unique combinations of coloured leg bands [25]. The database consisted of confirmed public sightings reported to BirdLife Australia and data from a standardized national monitoring programme based on 1367 sites throughout the breeding range [25]. We identified males individually through a combination of colour bands on the focal male or partner female ($n = 93$), nest location ($n = 68$), unique song attributes ($n = 21$) or a lack of other males nearby ($n = 42$) [25]. We recorded males' songs using a Sennheiser ME62/K6 microphone on a Telinga parabola and a Marantz PMD661 digital handheld recorder. We recorded captive-bred birds either shortly after their release into the wild in 2017 or in captivity in August 2019 and obtained historical song recordings of wild males (dated 1986–2011) from the Atlas of Living Australia and private sound collections. All individuals included in this study were at least 1 year old. See electronic supplementary material, texts S1 and S2 for further details of the captive breeding program and the historical song recordings, respectively.

## (c) Song classification

Wild male regent honeyeaters typically produce three distinct vocalizations: a soft, 'mewing' call; an alarm call consisting of a squawk and/or monosyllabic squeak; and a highly distinctive song, consisting of sub-chatter building to a crescendo of a

guttural warble produced with characteristic head-bobbing (electronic supplementary material, text S3).

The sightings database included 228 wild males identified since standardized contemporary monitoring commenced in 2015. We classified the songs of 146 of these males and were able to obtain quality recordings, defined as a high signal to noise ratio and no other background noises (so that all elements of the song were clearly visible in the spectrograms), of the songs of 47 of them. These males occurred in two different geographical regions: the Blue Mountains south of latitude −31.55 and the Northern Tablelands north of latitude −30.45 (figure 1*a*). We tested whether species-specific song types produced in these regions were significantly different using stepwise discriminant function analysis (DFA) of 15 song parameters (see electronic supplementary material, table S1 and data analysis section below). Eighteen of the 146 males, located throughout the contemporary range, failed to sing any species-specific songs and instead sang songs we considered similar to a different bird species (figures 1*a* and 2). We classified these birds as 'interspecific singers,' based either on visual similarities between spectrograms of the songs of interspecific singers and of the species whose songs we considered most similar ($n = 8$) or knowledge of the songs of the local avifauna in an experienced observer (RC, $n = 10$). We also obtained song recordings of 12 captive-reared males (three recorded one-week post-release in 2017 and nine recorded in captivity in 2019) and historic recordings of 14 wild males that were recorded prior to 2012 in the Blue Mountains. We tested for differences in the songs of the males of these five categories (Blue Mountains, Northern Tablelands, interspecific, captive-reared and historic) using the same DFA procedure described above. A single observer with 6 years' experience of monitoring regent honeyeaters (R.C.) recorded the songs of all but seven contemporary birds.

To quantify the repeatability of our song classifications, we asked seven professional ornithologists to assign blind a stratified, random sample of 20 songs to the contemporary song types and calculated the percentage agreement between our classification of each song and the classifications provided by the participants. We also asked each participant to identify the model species, if they thought that a recording was of an interspecific singer, and calculated the percentage agreement between our identification of the model species and that of the participants. See electronic supplementary material, text S4 for further information on the blind song classification procedure.

## (d) Data analysis

For all data analysis, we used R v. 3.4.3 [30] unless otherwise stated. We tested for spatial autocorrelation in the song type of contemporary wild males with correlograms of Moran's I, using package *ncf* v. 1.2–5 [31]. For acoustic analysis we first used Audacity v. 2.4.2 [32] to isolate songs in sound files and reduce background noise. We imported the trimmed .wav files of sufficient quality ($n = 73$ including contemporary wild, historic wild and captive birds) into *warbleR* v. 1.1.22 [33]. After restricting the frequency range to 0.5–5 kHz, we manually selected the start and end coordinates of each song, used *sig2noise* to increase the signal to noise ratio (type = 3) and *trackfreqs* to identify the spectral components of each spectrogram. We visually inspected spectrograms to ensure track frequencies selections were representative of the spectral components of each song and used *specan* to quantify 20 spectral attributes of each song (electronic supplementary material, table S1). We manually calculated a further three attributes based on visual and audial inspection of the recordings [11]: number of syllables, number of unique syllables and number of notes per syllable. We checked for pairwise correlation across all attributes using 'GGally' v. 1.4.0 [34], but no attributes showed a consistent strong correlation ($R > 0.5$ or $<-0.5$). We log-transformed modulation index, kurtosis, maximum dominant frequency and notes per syllable to fulfil normality assumptions.

We used JMP version 15.0 to conduct a discriminant function analysis (DFA) of songs by song type. We only included significant acoustic attributes ($n = 15$) in the final model via a backwards stepwise selection procedure (electronic supplementary material, table S1), and assessed the fit of the model by calculating the proportion of songs assigned to the correct song type.

To determine whether interspecific singers more frequently occurred at lower population density than species-specific singers, we calculated for each wild male the number of other males sighted in the same breeding season (June to January) within distance bands of less than 1 and less than 50 km. We considered these two spatio-temporal categories of ecological relevance to song learning, given the regent honeyeater's range size and capacity to undertake long-distance movements [23]. We used Mann–Whitney U-tests to look for a difference in the number of conspecifics located within both spatio-temporal windows, with the interspecific singer or not as the binomial response.

To quantify differences in song complexity between song types, we took 13 of the acoustic attributes that represented attributes of song complexity (electronic supplementary material, table S1) and fitted a general linear model of each attribute by song type using *lme4* v. 1.1-21 [35].

To assess the fitness costs of males' songs in the remaining wild population, we used logistic regression models with a binomial error structure and logit link function in *lme4*. For fitness analyses, we included in the dataset males whose songs we could not record, or could not record of sufficient quality for acoustic analysis, but could assign with high confidence to a song type in the field ($n = 105$) because they were clearly heard singing by an experienced observer (R.C.). We classified the songs of a further 63 males,

whose songs we could not assign to a song type as 'unknown' because we did not hear or record these males singing at the time they were detected, and not because their songs were intermediate between song types. The first model tested the effect of song type on whether a male was paired with a female or not. The second model tested whether song type affected the probability of paired males reaching the egg stage of nesting. The third model tested whether song type affected the probability of nesting males successfully fledging young. We then re-ran each model, reclassifying each male's song type binomially as 'regional cultural norm' or 'non-regional cultural norm'. We defined regional cultural norm as the typical Blue Mountains song in the Blue Mountains, and the Northern Tablelands song in the Northern Tablelands. We considered all other classified songs in each breeding area as 'non-regional cultural norm'.

To confirm that any fitness costs of the male song were associated with differences from the regional cultural norm and were not an artefact of song type classifications, we repeated the 'paired' and 'nested' logistic regression analyses, replacing the song type of each male located in the Blue Mountains for which we had a high-quality recording ($n = 34$) with the Mahalanobis distance of each males' song from the multivariate mean of the entire Blue Mountains population. We calculated the Mahalanobis distance of each male's song using *heplots* v. 1.3-5 [36]. Larger Mahalanobis distances represent greater song divergence from the multivariate mean [37]. For this analysis, we defined the regional cultural norm as the multivariate mean, rather than the most common song type. The small sample of quality recordings from the Northern Tablelands ($n = 7$) precluded us from repeating logistic regressions with Mahalanobis distances on the Northern Tablelands population.

To assess song repeatability, we used a Mantel test in *ade4* v. 1.7-15 [38] with 9999 permutations to compare the song similarity distance between repeat recordings of the same individuals to the average distance between all other males' songs. See the data availability section below for access to metadata detailing how we identified individuals and which individuals we included in each component of the statistical analysis.

## 3. Results

### (a) Frequency and distribution of song types

Discriminant function analysis revealed that males in the Blue Mountains and Northern Tablelands produced significantly different song types (figure 1*b*; electronic supplementary material, figure S1). These two song types were also readily audibly recognizable by experienced observers or through visual inspection of spectrograms (figure 2). In the Blue Mountains, 93 of 132 males sang the typical Blue Mountains song and this song type was not found elsewhere. In the Northern Tablelands, 17 of 22 males sang the Northern Tablelands song and 6 males sang this song type in the Blue Mountains, probably having dispersed there (figure 1*a*).

Some males produced song types that were atypical for their region (figures 1*a* and 2). Located exclusively in the Blue Mountains, 20 males produced a distinctive, abbreviated version of the typical Blue Mountains song (figures 1*a* and 2*d*). We therefore classified these birds' songs as their own song type—the 'clipped Blue Mountains' song. Located throughout the study area, eighteen males sang interspecific songs: five males' songs resembled songs of little wattlebird *Anthochaera chrysoptera*, four of noisy friarbird *Philemon corniculatus*, three of spiny-cheeked honeyeater *Acanthagenys rufogularis*, two of pied currawong *Streptera graculina*, and singles of eastern rosella *Platycercus eximius*, little friarbird *Philemon citreogularis*,

olive-backed oriole Oriolus sagittatus and black-faced cuckooshrike Coracina novaehollandiae (figures 1a and 2f–o). Using only a single song recording and with no field context (i.e. without any capacity to observe birds singing in the wild or in captivity), there was 89% agreement between our classification of song types and the classifications assigned by seven professional ornithologists. For interspecific singing regent honeyeaters, the participants identified the same model species as us in 79% of cases (electronic supplementary material, table S2).

Overall, 27% of contemporary wild males' songs differed from the regional cultural norm. Discriminant function analysis distinguished between these atypical songs and the region-typical songs with 93% accuracy (figure 1b; Wilks's $\lambda = 0.004$, approx. $F = 7.14$, $p < 0.001$, the first two canonicals accounted for 81% of total variance). A correlogram showed the song types of contemporary wild males were more similar to each other at distance classes between 10 and 110 km and less similar to each other between 130 and 200 km (figure 1a; electronic supplementary material, figure S2).

Recordings of wild males made between 1986 and 2011 in the Blue Mountains revealed that, historically, the predominant song type across the entire region was most similar to the Northern Tablelands song type (figure 1b). Songs produced by captive-bred males differed noticeably from those of all wild regent honeyeaters, both historic and contemporary (figures 1b and 2a–e).

## (b) Song repeatability

Individual regent honeyeaters consistently produced only one song type over time. Repeat recordings of the same individuals' songs were more similar to each other than to those of all other individuals (Mantel test, $n = 25$, Obs = 0.028, simulated $p = 0.015$). We recorded two colour-marked males in different years; one male produced the typical Blue Mountains song type in 2015 and 2017, and another produced the clipped Blue Mountains song type in 2016 and 2017 (electronic supplementary material, figure S3). Two males first recorded in the wild producing a typical Blue Mountains song in 2019 maintained this song type in captivity at least 18 months later, having been recruited to the captive population. We obtained repeat recordings of the songs of 21 individuals in the same season. We also observed a further five colour-marked males across years, whose songs we could not record but could consistently assign by ear to the typical Blue Mountains song type.

## (c) Population density and interspecific song learning

The production of atypical songs was predicted by population density. Males that sang other species' songs had significantly fewer conspecifics detected within 1 km and within 50 km in the same breeding season than males with species-specific songs (Mann–Whitney $U$-tests: 1 km, $W = 671$, $p = 0.001$; 50 km, $W = 488$, $p < 0.001$, figure 1c).

## (d) Song complexity

Species-specific regent honeyeater songs have simplified over time. General linear models revealed many song complexity metrics had negative $\beta$ effects for song types, relative to the pre-2012 Blue Mountains songs (figure 3; electronic supplementary material, figure S4 and table S3). Typical Blue Mountains singers had a lower maximum dominant frequency and lower spectral flatness. Clipped Blue Mountains singers had fewer syllables, fewer unique syllables and shorter song duration. Captive-bred birds had the least complex songs, being shorter with fewer syllables, fewer unique syllables and a flatter dominant frequency slope.

## (e) Fitness consequences of male song type

Production of atypical songs carried reproductive costs; males whose songs differed from the regional cultural norm were significantly less likely to be paired to a female (figure 4a–c and table 1). Specifically, males occurring in the Blue Mountains that sang clipped Blue Mountains, Northern Tablelands and interspecific songs were less likely to be paired than males singing the typical Blue Mountains song (figure 4a). Among paired males, those whose songs differed from the regional cultural norm were significantly less likely to initiate a nest that reached the egg stage. This effect was driven by a lower frequency of nesting in paired males with a clipped Blue Mountains song than in paired males with other song types (figure 4d,e). Across all contemporary males recorded in the Blue Mountains regardless of their song type, those whose songs were more divergent from the regional cultural norm, here defined as the Mahalanobis distance of each males' song from the multivariate mean, were less likely to be paired ($n = 34$, $\beta = -0.08$, s.e. = 0.04, $z = -2.13$, $p = 0.03$; figure 4c) but paired males were no less likely to nest ($n = 20$, $\beta = 0.002$, s.e. = 0.05, $z = 0.04$, $p = 0.97$; figure 4f). The song type of nesting birds did not affect their probability of fledging young (electronic supplementary material, table S4).

# 4. Discussion

Understanding how animal cultures are maintained and the conditions under which they are lost is important from both evolutionary and conservation perspectives [2,3,39]. Here, we provide rare evidence that, similar to the loss of human languages globally [9], a severe decline in population size and density is associated with substantial erosion of vocal culture in a wild animal population. We show for the first time that remaining individuals in such circumstances suffer significant fitness costs that may exacerbate already major threatening processes.

Regional song types clustered in multidimensional space, suggesting that like other birds and mammals, male regent honeyeaters learn vocalizations from nearby conspecifics and probably benefit from signalling group membership [6,40,41]. Sons do not learn songs from fathers, because adult males do not sing during the period that their offspring are resident on the natal territory and offspring are forced to disperse from natal areas before fathers recommence singing [25]. Independent juveniles must either co-occur with other singing males in the natal area, or they must disperse to locate and learn songs from other adult males in the landscape [42]. Given low breeding success rates and the very sparse distribution of breeding aggregations throughout the regent honeyeater's contemporary range [25] (often greater than 100 km apart; figure 1a), many juvenile males are probably unable to locate adult male tutors during their critical song learning period. Instead, these birds learn the songs of one of a wide range of other species that they may happen to associate with at that time (electronic supplementary material, text S8). Because interspecific singing is associated with lower fitness, we conclude interspecific song learning

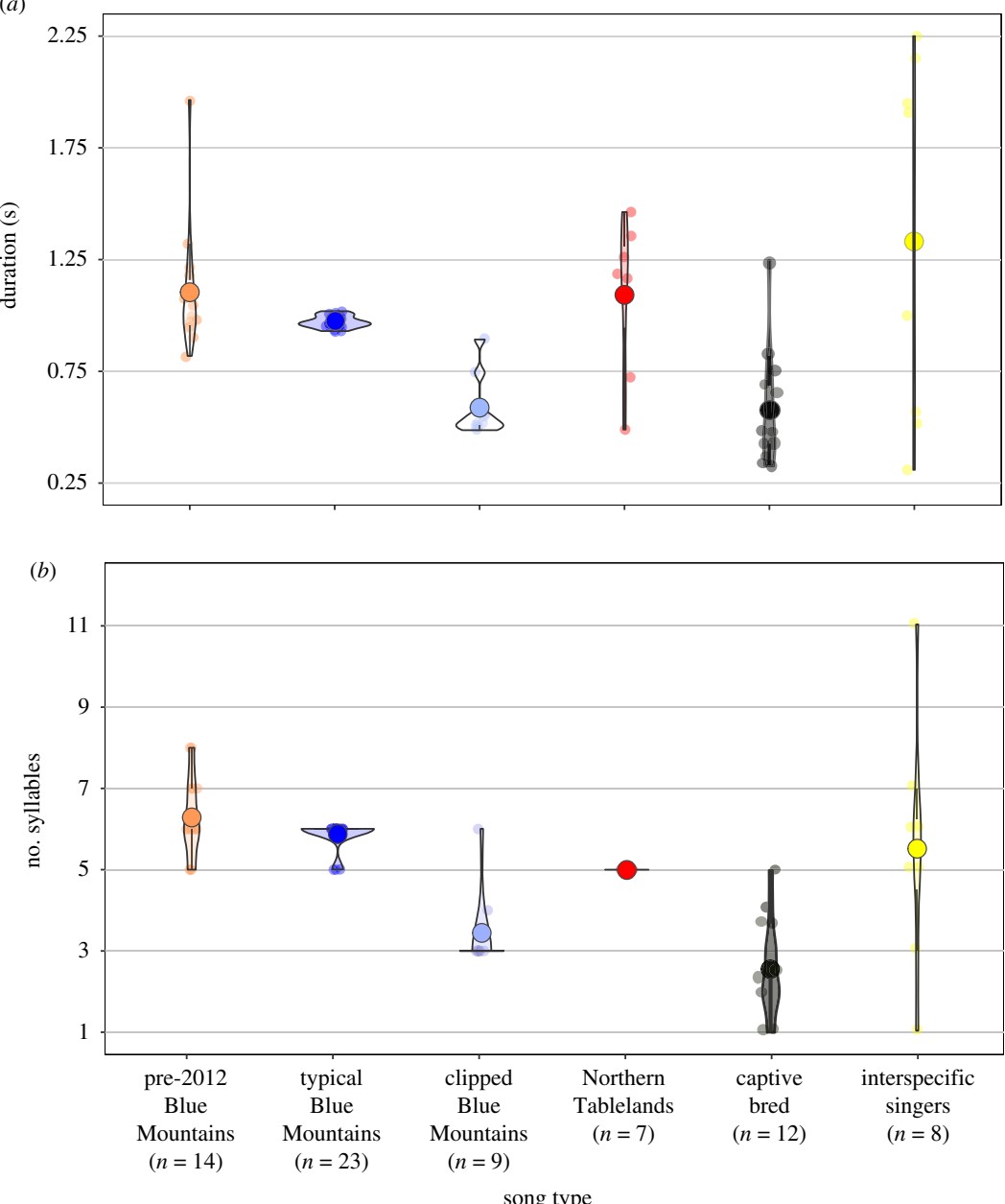

**Figure 3.** Differences in complexity of regent honeyeater songs by song type. Violin plots show (*a*) the duration of and (*b*) the number of syllables in regent honeyeater songs by song type. See electronic supplementary material, table S3 and figure S4 for details of other song complexity metrics. (Online version in colour.)

in regent honeyeaters is unlikely to represent vocal mimicry (vocal resemblance where a receiver specifically selects for resemblance to the model [43]). Other published examples of erroneous, interspecific vocal learning are typically limited to isolated cases, where one individual has learned another species's song [21,44]. Thus, interspecific vocal learning now appears to be occurring in regent honeyeaters at a frequency (12%) that is unprecedented in wild animal populations.

Recordings of captive-bred birds provide an insight into the mechanisms underpinning regent honeyeater song learning. Captive juveniles are typically crèched away from adults after fledging, meaning they do not associate with adult tutors during song learning [6]. Separation of captive juveniles from wild conspecifics over generations has led to a unique and simplified captive song culture, a process similar to song diversification in translocated populations [45]. An intriguing hypothesis to test is whether crèched captive juveniles are actually using each other as vocal tutors [46], leading to the fixation of a captive adult song culture similar to the developing songs of

wild juveniles. The distinct songs of captive-bred males could jeopardize their contribution to population recovery post-release, if wild females select against the captive song type in the same way they select against wild males whose songs differ from the cultural norm [47].

Consistent with vocal learning in other declining populations [14], we found that the complexity of species-specific regent honeyeater songs has declined over time. Male songs recorded between 1986 and 2011 were longer and had more syllables than contemporary songs. The typical Blue Mountains song occurred in the same geographical area as the abbreviated version of this song—the clipped Blue Mountains song. One possible explanation for this is that, as a result of low population density, a copying error by one individual was learned by other individuals who lacked alternative tutors in the vicinity of their territory, allowing the clipped song type to gradually spread through the population. Carry-over effects of early-life stress [48–50] may also be driving a decline in regent honeyeater song complexity.

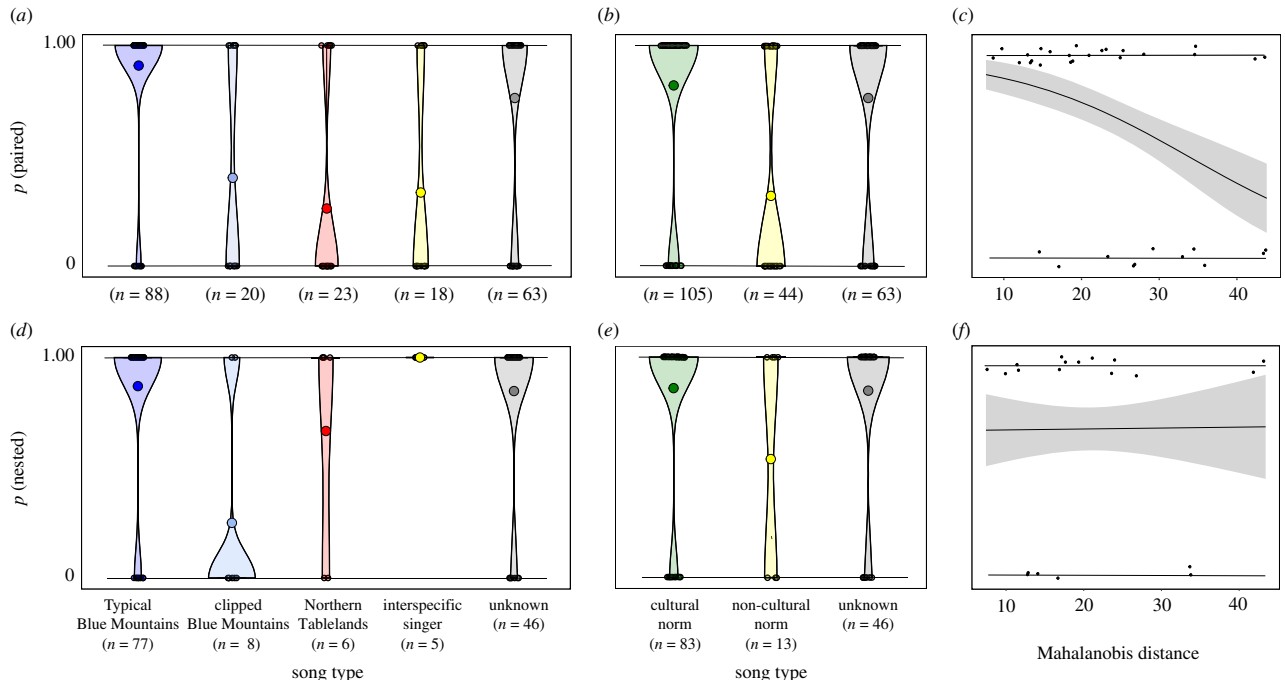

**Figure 4.** Fitness consequences of male song in contemporary wild regent honeyeaters. (*a*–*c*) Probability of males being paired to a female by (*a*) song type, (*b*) regional cultural norm and (*c*) Mahalanobis distance. (*d*–*f*) The probability of paired male regent honeyeaters securing a nesting territory and reaching the egg stage by (*d*) song type, (*e*) regional cultural norm and (*f*) Mahalanobis distance. Mahalanobis distance is the distance of each male's song from the multivariate centroid of the songs of males recorded between 2015 and 2019 in the Blue Mountains population only, regardless of song type. Line and shading in (*c*) and (*f*) denote model predictions and standard error from the logistic regression, respectively. (Online version in colour.)

**Table 1.** Effect of contemporary wild male regent honeyeater song type on their propensity to be paired to a partner female and of paired males to nest. Song type effects are relative to males singing the typical Blue Mountains song. Cultural norm effects are relative to males singing the regional cultural norm.

| response | test | level | β | se | Z | p |
|---|---|---|---|---|---|---|
| paired (n = 212) | song type | clipped Blue Mountains | −2.71 | 0.59 | −4.61 | <0.001** |
| | | Northern Tablelands | −3.34 | 0.60 | −5.55 | <0.001** |
| | | interspecific singer | −3.00 | 0.62 | −4.81 | <0.001** |
| | | unknown | −1.13 | 0.47 | −2.56 | 0.01* |
| | cultural norm | non-cultural norm | −2.27 | 0.41 | −5.53 | <0.001** |
| | | unknown | −0.35 | 0.39 | −0.89 | 0.37 |
| nested (n = 142) | song type | clipped Blue Mountains | −3.00 | 0.88 | −3.39 | <0.001** |
| | | Northern Tablelands | −1.21 | 0.93 | −1.30 | 0.19 |
| | | interspecific singer | 14.66 | 1073 | 0.01 | 0.99 |
| | | unknown | −0.18 | 0.53 | −0.35 | 0.73 |
| | cultural norm | non-cultural norm (1) | −1.62 | 0.64 | −2.55 | 0.01* |
| | | unknown (2) | −0.06 | 0.52 | −0.12 | 0.91 |

*denotes significance at *p* < 0.05 level.
**denotes significance at *p* < 0.01 level.

Considering the multiple costs of living in ever-smaller social groups [20], it is plausible that a temporal decline in regent honeyeater song complexity reflects the increasingly challenging conditions juveniles are experiencing in order to survive in the wild [29,51].

Five of the 18 interspecific singing males were paired and nested, proving that interspecific song learning does not present an absolute barrier to mate acquisition. Our data support a conceptual model whereby maladaptive song learning affects fitness, and potentially population growth rates, in a critical population density range [20] (electronic supplementary material, figure S5). Above this density range, maladaptive songs should be rare [17]. Below it, female choice may be limited to a single male regardless of his song type [52], concurrently reducing the strength of sexual selection on the male song [53].

Although we provide rare evidence of a fitness cost to maladaptive song learning, correlations between song type, population density and fitness do not necessarily imply causation [14]. Regent honeyeaters disperse away from breeding

grounds during summer to unknown areas [24], but we were still able to show that maladaptive song learning is negatively associated with the number of social associations an individual can obtain. Regent honeyeaters exhibit strong conspecific attraction, once roaming in flocks of hundreds [23]. However, at least four of the interspecific singers were more than 100 km from the nearest known male at the time they were detected. This suggests that regent honeyeaters now occur at population densities far below those at which they have evolved. Isolation during song learning presents parallel problems for mate-finding [52], since in many species a lack of females is greatest where population densities are lowest [54].

How cultural song erosion manifests itself in declining populations appears to depend on species-specific life-history attributes, such as mobility, social structure, range size and breeding biology [4,55]. Interspecific song learning has also been observed in endangered Hawaiian honeycreepers, but in these three species, song structures converged as the populations declined, possibly due to range contraction [11]. By contrast, population-level song diversity has increased in regent honeyeaters, through the emergence of interspecific singers, a song type unique to captive-bred males and spatial fragmentation of the species-specific song. Our observations of loss of vocal culture in regent honeyeaters draw parallels to cultural song change in humpback whales [56] and the loss of indigenous languages in humans [7]. Experiments in captivity that replicate a range of demographic scenarios could help improve our understanding of the process of cultural song erosion and its impacts on fitness and population growth [42].

Our study demonstrates that severe population decline is eroding culture in a wild animal population. The loss of culture is associated with individual fitness costs, but whether the loss of culture *contributes* to ongoing population decline remains an open question. Our findings in regent honeyeaters suggest that the loss of culture may be a precursor to extinction in declining populations that learn selected behaviours from conspecifics. Monitoring song cultures in wild populations may provide a useful indicator of population trajectory or threat status in species whose populations are otherwise very challenging to monitor directly.

Ethics. Research was conducted under New South Wales scientific licences #SL101580 and #SL101603 and Australian National University Animal Ethics permit #A2015/28.

Data accessibility. Data, sound files and an annotated R script are available from the Dryad Digital Repository: https://doi.org/10.5061/dryad.mkkwh70zj [57].

Authors' contributions. R.C. conceived the study, collected and analysed the data and wrote the manuscript. N.L. and L.Ran. helped with data analysis and, with R.H., D.S. and L.Ray., contributed to writing the manuscript. D.I. assisted data collection.

Competing interests. The authors declare no competing interests.

Funding. Research was funded by Cumnock Pty Ltd, Whithehaven Pty Ltd., the Mohamed Bin Zayed species conservation fund, New South Wales Department of Planning, Industry and Environment, Birding New South Wales, Hunter Bird Observers Club, Oatley Flora and Fauna and BirdLife Australia.

Acknowledgements. We thank Taronga Zoo, Joy Tripovic, OZCAM, Vicki Powys, Marc Anderson and James Lambert for providing song recordings. We thank Liam Murphy, Mick Roderick and all other fieldworkers that assisted with the national regent honeyeater monitoring program and seven professional ornithologists that undertook the blind classification of song types. We also thank two anonymous reviewers whose very detailed and thoughtful comments helped to greatly improve the manuscript. We acknowledge the traditional custodians of country upon which we conducted this research.

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
