## [Peer Review File · Proceedings of the Royal Society B: Biological Sciences]

Review History

RSPB-2020-2678.R0 (Original submission)

Review form: Reviewer 1

Recommendation

Major revision is needed (please make suggestions in comments)

Scientific importance: Is the manuscript an original and important contribution to its field?

Good

General interest: Is the paper of sufficient general interest?

Good

Quality of the paper: Is the overall quality of the paper suitable?

Acceptable

Is the length of the paper justified?

Yes

Should the paper be seen by a specialist statistical reviewer?

Yes

Do you have any concerns about statistical analyses in this paper? If so, please specify them explicitly in your report.

No

It is a condition of publication that authors make their supporting data, code and materials available - either as supplementary material or hosted in an external repository. Please rate, if applicable, the supporting data on the following criteria.

Is it accessible?

Yes

Is it clear?

Yes

Is it adequate?

Yes

Do you have any ethical concerns with this paper?

No

Comments to the Author

Review of Crates et al. Loss of vocal culture has fitness costs in a critically endangered songbird

This ambitious manuscript describes the results of research attempting to link population declines of an endangered songbird with cultural loss of sexual signals and then further show that this cultural loss in turn has fitness consequences. The authors document a number of male birds that sing atypical songs and show that these males tend to be more isolated from other birds and less likely to be paired or have nests, suggesting a relationship between these three variables. I think this subject is of broad interest and I found the manuscript to be generally well-written. Unfortunately, I think there are some issues with the manuscript as well.

The first issue is that the authors overstretch the interpretation of their data. Ultimately, the authors have demonstrated correlations between three traits (population size, song type, and fitness), yet the manuscript implies causation (reduction in population size \square poor learning opportunities \square reduced fitness). This is certainly one logical interpretation, but it very well could be the case that the causal relationship goes in a different direction, is more complex, or has a different mechanism altogether. To me, terms like "linked to" imply causation, which does not seem appropriate here.

Second, while the manuscript is well written, it is often missing important information, which sometimes makes it difficult to evaluate the details of the methods. One really great thing about this paper is that it is really ambitious in terms of how many topics and types of data it synthesizes. However I think that this breadth also makes it challenging to present all of the relevant details of all parts of the study, and this is especially true in a relatively short form like Proceedings. I think more clarity is needed in many of the experimental details, for example how songs were evaluated, whether birds were banded, which birds were included in different samples, etc. (more detailed comments below by line number).

Last, I had a few concerns about the methodological approach and experimental logic. As I mentioned above, details are not always clear in the manuscript, so I'm not entirely sure if I am always interpreting the methods correctly, but I think that a couple of the issues listed below could be quite important, especially those dealing with how song types are classified, which birds are included, and whether it is appropriate to compare the historical songs with contemporary songs because of the date range of historical songs and the geographic range of contemporary

songs.

Below I provide more details about these concerns, and other more minor issues, by line number:

L19 (and others): The authors do well to explicitly point out that correlation does not imply causation in the discussion section, but the rest of the manuscript is written as though there is a causal link between the variables. Terms like “linked to” imply causation, but this cannot actually be inferred from the data. I think the authors need to tone down their wording so as not to imply causation between correlated variables.

L34. While not commonly studied, there are at least a few examples of avian vocal culture changing in small populations. A few studies that come to mind are below, though there are probably others.

Laiolo, P., Vögeli, M., Serrano, D. and Tella, J.L., 2008. Song diversity predicts the viability of fragmented bird populations. *PLoS One*, 3(3), p.e1822.

Ortega, Y.K., Benson, A. and Greene, E., 2014. Invasive plant erodes local song diversity in a migratory passerine. *Ecology*, 95(2), pp.458-465.

(I note that this one is cited in the discussion). Valderrama, S.V., Molles, L.E. and Waas, J.R., 2013. Effects of population size on singing behavior of a rare duetting songbird. *Conservation Biology*, 27(1), pp.210-218.

Martínez, T.M. and Logue, D.M., 2020. Conservation practices and the formation of vocal dialects in the endangered Puerto Rican parrot, *Amazona vittata*. *Animal Behaviour*, 166, pp.261-271.

86: perhaps a bit more info about the breeding program would be useful here. For example, how many birds are released, and what proportion of the population is this? What are the rearing conditions and song learning opportunities in captivity? Etc.

77/80/95: are all birds included in the database uniquely marked? This is not reported, but it seems important to ensure that all of the birds included in the study are unique individuals and, given that this species is nomadic, this seems difficult to be sure of unless all birds are banded.

96: I'm assuming that a given male of this species sings only a single song type, rather than a repertoire. Is this correct? Please state this explicitly and provide a citation.

110: please clarify that these 7 songs were not unequally distributed across the sampling populations.

95-115: I'm a bit confused about the sample sizes reported here. Early in this section it is reported that out of 251 males, there were 161 males who sang yielding 47 quality recordings, but later on it is reported that there are 73 wav files that were of high quality to analyze. I'm especially confused about the 47 vs 73 discrepancy. I'm also confused about the additional birds who were studied but not recorded – were assessments made about these based simply on how they sounded to the observer or were these birds excluded from the analyses? The former seems quite problematic, but if the later I don't really understand why they are included in the manuscript.

Having now looked at the figures, I'm coming back to this comment as I would think it would be quite difficult to discern some of these song types from ear alone – some types are quite similar to the interspecific songs, for example. It is possible that the authors can do this, but I think including some more information about how the authors have ensured that these field assessments (if used) are reliable and repeatable would be important to include.

144: why could these males' songs not be classified to a song type?

241/ Fig 3: The authors test the idea that songs have become less complex over time by examining the pre 2012 songs with the current songs. One concern is that there are only 14 songs recorded in the entire period of 1986-2011. That seems like a pretty small sample for judging historical patterns. Also, if there are statistical changes in the songs from 2012-2018 (6 years), then is it really appropriate to lump together songs from 1986-2011 (25 years) and assume that they have not changed during this time? No information is provided, that I can find, about when in this period these songs were recorded, but given the changes proposed in more recent songs, this seems like critical information to include.

Next, I'm not really clear about the expected findings for this analysis given that all of the historical songs were recorded in one population in the Blue Mountains. Above caveats aside, I'm not sure this analysis is an appropriate way to test the question about whether songs have become less complex over time. I can see how comparing historical Blue Mountain songs with contemporary Blue Mountain songs would be interesting and address this question, but why should there be a relationship between historical Blue Mountain songs and contemporary songs from other populations? Is there a reason to think that all birds used to sing the Blue Mountain dialect no matter where they lived? Otherwise, there are too many variables changing between these samples and while it is fine to note that some dialects/populations are more complex than others it is not appropriate to then equate this with a loss in complexity over time as implied.

295: The authors emphasize the importance of a critical learning period here and at other places in the manuscript. Is there evidence that this species has a critical period or when this period takes place?

333: Interesting that the winter grounds of the birds are unknown. I wonder if birds could be associating with other individuals at this time, potentially providing opportunities for mate acquisition outside the breeding season or even song learning? As the authors note, the song learning must take place after dispersal from the natal territory, so could some of the song learning take place at these, unknown, locations?

Review form: Reviewer 2

Recommendation

Accept with minor revision (please list in comments)

Scientific importance: Is the manuscript an original and important contribution to its field?

Excellent

General interest: Is the paper of sufficient general interest?

Excellent

Quality of the paper: Is the overall quality of the paper suitable?

Good

Is the length of the paper justified?

Yes

Should the paper be seen by a specialist statistical reviewer?

No

Do you have any concerns about statistical analyses in this paper? If so, please specify them explicitly in your report.

No

It is a condition of publication that authors make their supporting data, code and materials available - either as supplementary material or hosted in an external repository. Please rate, if applicable, the supporting data on the following criteria.

Is it accessible?

Yes

Is it clear?

Yes

Is it adequate?

No

Do you have any ethical concerns with this paper?

No

Comments to the Author

This study documents the song variation shown by male regent honeyeaters, a critically endangered songbird, in relation to their population density – at low densities, males are more likely to sing atypical songs (either not the commonest song of the area, or a song resembling that of another species), and such males are less likely to pair and build a nest. In addition, songs of captive-raised males are even more atypical. Given that song is important for reproduction in all songbirds, these results likely have conservation consequences for the study species. The study also has important wider implications for a least two large areas of interest. First, the UNEP has recognized the potential impact of culture in conservation and called for more evidence – this study is a very important example of such impact and as such is likely to be widely cited in the conservation literature across all taxa. Second, as a relatively rare example of song acquisition in the wild this study adds important information relevant to song learning. Lab studies predominate in the song learning literature and such studies invariably design out social interaction and focus on early life. Yet studies in the wild generally point to the importance of social interaction and have shown learning in most species extends well beyond the nestling phase. This study's well-documented examples of relatively common singing of other species' songs is particularly welcome, since most other information on this behaviour is usually in the form of a short note with little or no supporting acoustic information (usually the birds were thought to sound similar, but were not recorded).

In summary, this is an important piece of research that is likely to be commonly cited in a number of areas of current research and it has practical conservation implications. However, the clarity of writing and the level of detail can be increased, and this should help ensure that the study is as widely read and cited as it deserves to be.

General comments:

Interspecific and captive songs. Your study is unique in combining such songs with information on songs of contemporary singers in the wild and, as you point out around L.299-302, reports of interspecific singing are usually single individuals, so the level you report in honeyeaters is unprecedented. To make the most of the insights these males' songs could provide needs some more detail to be added to the text.

For the captive facility the detail should include any factor that could influence song development (location, housing conditions, in flocks, alone but in earshot, presence of other species etc) and the source of original captive breeders etc.

For interspecific song, detail on the following would allow the reader to better assess the reported similarities at a number of places in the text:

- in section (c) of Methods report where the songs of other species used for comparison

came from (the memory of an experienced observer is an OK answer)

- L.180 in Results you should report the result you found (assessed visual / acoustic similarity of spectrogram / heard in field respectively between study species and another species) rather than an interpretation (“that had learned the songs of”).
- L.216 As previous comment, a form of words such as ‘Images of the other species with song most similar to honeyeater ...’
- L.235-6 Add location of xeno-canto song, or perhaps how far from the particular male honeyeater. As your study and many others have shown how variable even “species typical” songs are, please state how these songs were chosen. Did they particularly look like / sound like that specific honeyeater song? This level of detail is important to allow the reader to assess the significance of the similarity you are drawing attention to.
- L.296 “other species that they may happen to associate with” suggests that you have data on the presence of the other species with honeyeaters and it would help interpretation to know what that is, with direct field observations giving more weight to the interpretation than an overlapping geographical range known from the literature.

Qualitative v. quantitative measures of song similarity. These differ considerably in the level of detailed methodology you report; with access to the song measures data set and the detail on quantitative methods used, it would be possible to replicate your analyses. Two aspects of the qualitative analyses mean that replication would not be possible. The first is the data set, given the issue you have had with uploading zip files, could you consider depositing the recordings you analysed in an online archive (best if maintained by one of the big sound archives like Cornell). The second is that there is no comparable detail on the qualitative analysis to the R version, package (in ms) and scripts (in Supplementary data).

Relevant detail could be added throughout the ms (or as Supplementary material if space is too limited). Two specific points in the text where detail is needed are noted below (L.143, L.170). Simply expanding terms like “remained consistent” at L.199 and “field-validated” etc L.204 would be insightful.

“Tutor”. I strongly recommend replacing this term in captions for Figures and Tables, the third column of Fig.2, and at most places in the text (the exception would be when reporting lab learning experiments that have used the term). The reason is that you are reporting similarities between songs, either seen on spectrograms or heard, and this is very different from a lab song learning experiment in which the learner is presented with a singing tutor male. Even in the lab learning case it could be argued that the term is inappropriate but using it for your results risks obscuring or confusing those results. The main issue is that the term describes only one way in which the similarity between songs could have arisen. Males A and B could sing similar songs (to our perceptions) for other reasons than A learned from B, including B learning from A, both learning from C and chance. In lab experiments it is usually possible to exclude alternatives, but in field studies this is rarely the case. It doesn’t make your results any less interesting or important to report them as similarities, assessed either qualitatively or quantitatively and it does allow the full range of possible explanations for the similarities to be considered and discussed. Spectrograms: Settings, axis scaling and labelling. This detail is important to allow any reader to easily make the comparisons / note the similarities highlighted by the authors.

- The setting used by the software to produce the spectrograms is important detail to include in the Methods (sampling rate, window setting etc) because it affects both the visual appearance of the spectrogram and may affect quantitative analysis (especially if measurements are taken directly from the spectrogram).
- Similarly, the axes must allow direct comparison by having the same scaling. The time axis of Figure 2 and Fig S3 vary within the figures and should be standardized.
- The units of the time axis in Fig.2 should be changed to “Time (s)” from “Time (s or ms)”. The same time unit should apply to all parts of Fig.S3.

Frequency range and song complexity. Can you reference a study that has used frequency range as an indication of song complexity (e.g. L. 244) or provide a short explanation?

Specific comments (most are suggestions to increase clarity and reduce word count to allow inclusion of necessary extra detail):

Title. Consider replacing “has” with the more neutral ‘and’

L.12 This sentence could perhaps be reworded. The obvious conservation issue is population density decline, the question is whether cultural loss increases or otherwise exacerbates the decline. Perhaps ‘Declines in population density could be exacerbated by culture loss, thereby linking culture to conservation.’ You make this point in this sort of wording at L.36 and L.284

L.15-18 These sentences could be reworded. Partly because this is the only place “fidelity” is mentioned in the ms and partly because this is where a clear summary of the main findings is usual. ‘Wild males at low population density tend to sing atypical songs, either unusual for the area (27%) or resembling other species’ songs (16%).’

L.18- 19 This sentence combines two themes and splits the link between atypical song production and fitness. You could move up the L.19-22 result, perhaps expressed as ‘Males singing atypical songs were less likely to pair and nest than males that sang the regional cultural norm.’

and follow with the L.18-19 sentence, split into 2 sentences.

These changes would lead very neatly into the penultimate sentence – perhaps starting ‘We therefore ...’

L.19-22 see previous comment

L.23-24 Consider strengthening the concluding sentence by adding ‘and therefore provide a useful conservation indicator.’

L.34 Perhaps rephrase part of sentence to acknowledge that there is some evidence for vocal culture degradation in species other than humans, examples include Holland et al 1996, *J. Avian Biology* 27, 47-55; Osiejuk & Ratynska 2003, *Folia Zool* 52, 275-286.

L.47 It would help many readers to have an idea in km of what constitutes long distances in this context, perhaps replace “long distances” with ‘100s km’ or ‘1000s km’

L.52-54 It is perhaps worth making the point that this is probably not unusual in the wild, as several species have been documented as learning songs from their territorial neighbours rather than fathers (there are several references in the section on Song learning in birds in Garland & McGregor 2020 *Frontiers in Psychology* doi: 10.3389/fpsyg.2020.544929), so it is likely to be a general issue in song learning in declining populations.

L.63 As at L.47 it would be help many readers to replace “vast” with a numerical indication of range e.g. 1000s km²

L.80 The part of the sentence after “and” isn’t really a sequitur (unique bands don’t sex an individual). It seems likely that individuals were sexed in the hand during banding (because individual identification is dealt with in lines 82 onwards), so this part of the sentence could be replaced with ‘and in the hand during marking with unique combinations of coloured leg bands’.

L.86 See general comment on captive birds above. Given the importance of location to this study, at this point in the ms it would be helpful to note where captive birds were released.

L.92 ff It is not clear to me from the Supp. file S2 why the second sentence cannot be combined with the first. Could “...squeak; and a song etc” be replaced with ‘... squeak; and a highly distinctive song, consisting of ... warble (Supplementary file S1) produced with characteristic head-bobbing (Supplementary file S2).’

L.97 (also 115) Please define / expand “quality” so that the reader understands what you consider to be high quality (e.g., lack of background noise, high signal to noise ratio, recorded from within ?20m etc). The reason that this is important is to allow the reader to judge whether this selection process could have biased the data used in analysis (see the STRANGE framework on sampling bias Webster & Rutz 2020 *Nature* 582, 337-).

L.101-104 See request above for more detail on this aspect of the study. Perhaps the phrase “failed to sing any species-specific songs and instead produced the song of a different bird species” could be reworded to include some of this detail, for example ‘were heard to sing songs that resembled another species. Eight of these ‘interspecific singers’ were recorded.’

L.109-110 Perhaps this sentence could be made clearer, as “obtained” may refer to recordings made by the experienced observer (where the observer’s experience is relevant) and by others (where the first part of the sentence is not relevant).

- L.117 It seems unlikely that your aim was “to reduce signal to noise ratio”, more likely you aimed ‘to increase signal to noise ratio’?
- L.118 and ff The term “sonogram” is used in this line at in a table heading (S1), whereas spectrogram is used at L.170 and in Figure 2 caption. Spectrogram is the more standard usage and would be better used throughout the ms for consistency.
- L.134 Remove “vast” as it is an unreferenced adjective and the detail will now be available after change at L.47
- L.137 Why 14 attributes here when 15 were used previously (L.127)?
- L.143 Explain how they could be assigned with high confidence – perhaps they had been clearly / routinely heard by the experienced observer? This also relates to the categorization of “unknown” song types.
- L.149 Replace “... males’ ...” with ‘... male’s ...’
- L.160 Please say how small. E.g., ‘The small (5) sample ...’
- L. 164 Perhaps expand “distance” to ‘song similarity distance’ to prevent confusion with geographic distance.
- L.170 Please expand detail. Were songs “readily recognizable” by the experienced observer? By all the authors?
- L.171-173, 175 & 179, 185 This is a key result, but difficult to understand in its current form even read alongside Fig1A and with reference back to male totals in the Methods. Please reword using numbers of males singing the variant and the number of all males in that area. I can’t work these numbers out from Figures + Methods. Something like the following would be clearer ‘In the Blue Mountains ?? of ?? males sang the typical Blue Mountains song and this song was not found elsewhere. In the Northern Tablelands ?? of ?? males, sang the Northern Tablelands song and 6 males sang it elsewhere.’
- L.180 See general comment above on song similarity as evidence of learning. An accurate and neutral reporting of the data would be ‘Throughout the study area, 17? males sang interspecific songs: 4 resembled songs of noisy friarbird, 5 little wattlebird,’ etc etc
- L.193 “significantly” implies a supporting statistical analysis, replace with ‘noticeably’
- L.197 To be clearer could replace “Repeat recordings ...” with ‘Recordings of the same individual within a year were ...’
- L.199 “remained consistent” see general comment above
- L.203 Which males are “the other 21 individuals in the same season” referring to? Are these the captive population? This detail needs to be clarified and more obviously related to the surrounding text.
- L.204 “field-validated” see general comment above
- L.224 Can combine sentences “... with a circle. The size ... corresponds” as ‘... with a circle corresponding’
- L.227 Can remove “spatial windows of <”
- L.216 and throughout captions, ensure label e.g., (1), (A) occurs before the item they refer to.
- L.232, 233 Also for subsection labels (A-E) Species-specific etc
- L.237 see general comment above on axis scaling for comparisons
- L.275 Add information on what the line with shading indicates in (C) and (F)
- L.325 Add of total interspecific (Five out of 18 of the interspecific singing ...) to aid interpretation.
- L.343 Is it known whether Hawaiian honeycreepers became fragmented and widespread (like regent honeyeaters) or concentrated into a smaller area of suitable habitats? If the latter, this could explain song convergence.
- L.353-355 Perhaps replace, or add, a comment on the practical conservation value – indications of a loss of culture may be a useful conservation tool to establish the level of threat faced by declining populations?

Decision letter (RSPB-2020-2678.R0)

29-Dec-2020

Dear Mr Crates:

I am writing to inform you that your manuscript RSPB-2020-2678 entitled "Loss of vocal culture has fitness costs in a critically endangered songbird" has, in its current form, been rejected for publication in Proceedings B.

This action has been taken on the advice of referees, who have recommended that substantial revisions are necessary. With this in mind we would be happy to consider a resubmission, provided the comments of the referees are fully addressed. However please note that this is not a provisional acceptance.

Sincerely,
Dr Sasha Dall
<mailto:proceedingsb@royalsociety.org>

Associate Editor
Board Member: 1

Comments to Author:

Both reviewers feel that this is an important study, but that there is substantial room for improvement, particularly in the way that the data is presented. Also some claims are made that could be moderated somewhat, and their importance made more accessible.

One reviewer feels that some of the methods used require more information and more explanation - these could be provided at length in the SI, as well as explained more clearly in the main text.

This reviewer also raises important issues in interpretation (esp in terms of song classification, and the analysis of historical as well as contemporary songs together, and in potentially overstating your conclusions (particularly in terms of inferring cause from associations) that you should address in any new version.

Reviewer(s)' Comments to Author:

Referee: 1

Comments to the Author(s)

Review of Crates et al. Loss of vocal culture has fitness costs in a critically endangered songbird

This ambitious manuscript describes the results of research attempting to link population declines of an endangered songbird with cultural loss of sexual signals and then further show that this cultural loss in turn has fitness consequences. The authors document a number of male birds that sing atypical songs and show that these males tend to be more isolated from other birds and less likely to be paired or have nests, suggesting a relationship between these three variables. I think this subject is of broad interest and I found the manuscript to be generally well-written. Unfortunately, I think there are some issues with the manuscript as well.

The first issue is that the authors overstretch the interpretation of their data. Ultimately, the authors have demonstrated correlations between three traits (population size, song type, and fitness), yet the manuscript implies causation (reduction in population size \square poor learning opportunities \square reduced fitness). This is certainly one logical interpretation, but it very well could be the case that the causal relationship goes in a different direction, is more complex, or has a different mechanism altogether. To me, terms like "linked to" imply causation, which does not seem appropriate here.

Second, while the manuscript is well written, it is often missing important information, which sometimes makes it difficult to evaluate the details of the methods. One really great thing about this paper is that it is really ambitious in terms of how many topics and types of data it synthesizes. However I think that this breadth also makes it challenging to present all of the relevant details of all parts of the study, and this is especially true in a relatively short form like Proceedings. I think more clarity is needed in many of the experimental details, for example how songs were evaluated, whether birds were banded, which birds were included in different samples, etc. (more detailed comments below by line number).

Last, I had a few concerns about the methodological approach and experimental logic. As I mentioned above, details are not always clear in the manuscript, so I'm not entirely sure if I am always interpreting the methods correctly, but I think that a couple of the issues listed below could be quite important, especially those dealing with how song types are classified, which birds are included, and whether it is appropriate to compare the historical songs with contemporary songs because of the date range of historical songs and the geographic range of contemporary songs.

Below I provide more details about these concerns, and other more minor issues, by line number:

L19 (and others): The authors do well to explicitly point out that correlation does not imply causation in the discussion section, but the rest of the manuscript is written as though there is a causal link between the variables. Terms like "linked to" imply causation, but this cannot actually be inferred from the data. I think the authors need to tone down their wording so as not to imply causation between correlated variables.

L34. While not commonly studied, there are at least a few examples of avian vocal culture changing in small populations. A few studies that come to mind are below, though there are probably others.

Laiolo, P., Vögeli, M., Serrano, D. and Tella, J.L., 2008. Song diversity predicts the viability of fragmented bird populations. *PLoS One*, 3(3), p.e1822.

Ortega, Y.K., Benson, A. and Greene, E., 2014. Invasive plant erodes local song diversity in a migratory passerine. *Ecology*, 95(2), pp.458-465.

(I note that this one is cited in the discussion). Valderrama, S.V., Molles, L.E. and Waas, J.R., 2013. Effects of population size on singing behavior of a rare duetting songbird. *Conservation Biology*, 27(1), pp.210-218.

Martínez, T.M. and Logue, D.M., 2020. Conservation practices and the formation of vocal dialects in the endangered Puerto Rican parrot, *Amazona vittata*. *Animal Behaviour*, 166, pp.261-271.

86: perhaps a bit more info about the breeding program would be useful here. For example, how many birds are released, and what proportion of the population is this? What are the rearing conditions and song learning opportunities in captivity? Etc.

77/80/95: are all birds included in the database uniquely marked? This is not reported, but it seems important to ensure that all of the birds included in the study are unique individuals and, given that this species is nomadic, this seems difficult to be sure of unless all birds are banded.

96: I'm assuming that a given male of this species sings only a single song type, rather than a repertoire. Is this correct? Please state this explicitly and provide a citation.

110: please clarify that these 7 songs were not unequally distributed across the sampling populations.

95-115: I'm a bit confused about the sample sizes reported here. Early in this section it is reported that out of 251 males, there were 161 males who sang yielding 47 quality recordings, but later on it is reported that there are 73 wav files that were of high quality to analyze. I'm especially confused about the 47 vs 73 discrepancy. I'm also confused about the additional birds who were studied but not recorded – were assessments made about these based simply on how they sounded to the observer or were these birds excluded from the analyses? The former seems quite problematic, but if the later I don't really understand why they are included in the manuscript.

Having now looked at the figures, I'm coming back to this comment as I would think it would be quite difficult to discern some of these song types from ear alone – some types are quite similar to the interspecific songs, for example. It is possible that the authors can do this, but I think including some more information about how the authors have ensured that these field assessments (if used) are reliable and repeatable would be important to include.

144: why could these males' songs not be classified to a song type?

241/ Fig 3: The authors test the idea that songs have become less complex over time by examining the pre 2012 songs with the current songs. One concern is that there are only 14 songs recorded in the entire period of 1986-2011. That seems like a pretty small sample for judging historical patterns. Also, if there are statistical changes in the songs from 2012-2018 (6 years), then is it really appropriate to lump together songs from 1986-2011 (25 years) and assume that they have not changed during this time? No information is provided, that I can find, about when in this period these songs were recorded, but given the changes proposed in more recent songs, this seems like critical information to include.

Next, I'm not really clear about the expected findings for this analysis given that all of the historical songs were recorded in one population in the Blue Mountains. Above caveats aside, I'm not sure this analysis is an appropriate way to test the question about whether songs have become less complex over time. I can see how comparing historical Blue Mountain songs with

contemporary Blue Mountain songs would be interesting and address this question, but why should there be a relationship between historical Blue Mountain songs and contemporary songs from other populations? Is there a reason to think that all birds used to sing the Blue Mountain dialect no matter where they lived? Otherwise, there are too many variables changing between these samples and while it is fine to note that some dialects/populations are more complex than others it is not appropriate to then equate this with a loss in complexity over time as implied.

295: The authors emphasize the importance of a critical learning period here and at other places in the manuscript. Is there evidence that this species has a critical period or when this period takes place?

333: Interesting that the winter grounds of the birds are unknown. I wonder if birds could be associating with other individuals at this time, potentially providing opportunities for mate acquisition outside the breeding season or even song learning? As the authors note, the song learning must take place after dispersal from the natal territory, so could some of the song learning take place at these, unknown, locations?

Referee: 2

Comments to the Author(s)

This study documents the song variation shown by male regent honeyeaters, a critically endangered songbird, in relation to their population density – at low densities, males are more likely to sing atypical songs (either not the commonest song of the area, or a song resembling that of another species), and such males are less likely to pair and build a nest. In addition, songs of captive-raised males are even more atypical. Given that song is important for reproduction in all songbirds, these results likely have conservation consequences for the study species. The study also has important wider implications for a least two large areas of interest. First, the UNEP has recognized the potential impact of culture in conservation and called for more evidence – this study is a very important example of such impact and as such is likely to be widely cited in the conservation literature across all taxa. Second, as a relatively rare example of song acquisition in the wild this study adds important information relevant to song learning. Lab studies predominate in the song learning literature and such studies invariably design out social interaction and focus on early life. Yet studies in the wild generally point to the importance of social interaction and have shown learning in most species extends well beyond the nestling phase. This study's well-documented examples of relatively common singing of other species' songs is particularly welcome, since most other information on this behaviour is usually in the form of a short note with little or no supporting acoustic information (usually the birds were thought to sound similar, but were not recorded).

In summary, this is an important piece of research that is likely to be commonly cited in a number of areas of current research and it has practical conservation implications. However, the clarity of writing and the level of detail can be increased, and this should help ensure that the study is as widely read and cited as it deserves to be.

General comments:

Interspecific and captive songs. Your study is unique in combining such songs with information on songs of contemporary singers in the wild and, as you point out around L.299-302, reports of interspecific singing are usually single individuals, so the level you report in honeyeaters is unprecedented. To make the most of the insights these males' songs could provide needs some more detail to be added to the text.

For the captive facility the detail should include any factor that could influence song development (location, housing conditions, in flocks, alone but in earshot, presence of other species etc) and the source of original captive breeders etc.

For interspecific song, detail on the following would allow the reader to better assess the reported similarities at a number of places in the text:

- in section (c) of Methods report where the songs of other species used for comparison came from (the memory of an experienced observer is an OK answer)
- L.180 in Results you should report the result you found (assessed visual / acoustic similarity of spectrogram / heard in field respectively between study species and another species) rather than an interpretation (“that had learned the songs of”).
- L.216 As previous comment, a form of words such as ‘Images of the other species with song most similar to honeyeater ...’
- L.235-6 Add location of xeno-canto song, or perhaps how far from the particular male honeyeater. As your study and many others have shown how variable even “species typical” songs are, please state how these songs were chosen. Did they particularly look like / sound like that specific honeyeater song? This level of detail is important to allow the reader to assess the significance of the similarity you are drawing attention to.
- L.296 “other species that they may happen to associate with” suggests that you have data on the presence of the other species with honeyeaters and it would help interpretation to know what that is, with direct field observations giving more weight to the interpretation than an overlapping geographical range known from the literature.

Qualitative v. quantitative measures of song similarity. These differ considerably in the level of detailed methodology you report; with access to the song measures data set and the detail on quantitative methods used, it would be possible to replicate your analyses. Two aspects of the qualitative analyses mean that replication would not be possible. The first is the data set, given the issue you have had with uploading zip files, could you consider depositing the recordings you analysed in an online archive (best if maintained by one of the big sound archives like Cornell). The second is that there is no comparable detail on the qualitative analysis to the R version, package (in ms) and scripts (in Supplementary data).

Relevant detail could be added throughout the ms (or as Supplementary material if space is too limited). Two specific points in the text where detail is needed are noted below (L.143, L.170). Simply expanding terms like “remained consistent” at L.199 and “field-validated” etc L.204 would be insightful.

“Tutor”. I strongly recommend replacing this term in captions for Figures and Tables, the third column of Fig.2, and at most places in the text (the exception would be when reporting lab learning experiments that have used the term). The reason is that you are reporting similarities between songs, either seen on spectrograms or heard, and this is very different from a lab song learning experiment in which the learner is presented with a singing tutor male. Even in the lab learning case it could be argued that the term is inappropriate but using it for your results risks obscuring or confusing those results. The main issue is that the term describes only one way in which the similarity between songs could have arisen. Males A and B could sing similar songs (to our perceptions) for other reasons than A learned from B, including B learning from A, both learning from C and chance. In lab experiments it is usually possible to exclude alternatives, but in field studies this is rarely the case. It doesn’t make your results any less interesting or important to report them as similarities, assessed either qualitatively or quantitatively and it does allow the full range of possible explanations for the similarities to be considered and discussed. Spectrograms: Settings, axis scaling and labelling. This detail is important to allow any reader to easily make the comparisons / note the similarities highlighted by the authors.

- The setting used by the software to produce the spectrograms is important detail to include in the Methods (sampling rate, window setting etc) because it affects both the visual appearance of the spectrogram and may affect quantitative analysis (especially if measurements are taken directly from the spectrogram).
- Similarly, the axes must allow direct comparison by having the same scaling. The time axis of Figure 2 and Fig S3 vary within the figures and should be standardized.
- The units of the time axis in Fig.2 should be changed to ‘Time (s)’ from “Time (s or ms)”. The same time unit should apply to all parts of Fig.S3.

Frequency range and song complexity. Can you reference a study that has used frequency range as an indication of song complexity (e.g. L. 244) or provide a short explanation?

Specific comments (most are suggestions to increase clarity and reduce word count to allow inclusion of necessary extra detail):

Title. Consider replacing “has” with the more neutral ‘and’

L.12 This sentence could perhaps be reworded. The obvious conservation issue is population density decline, the question is whether cultural loss increases or otherwise exacerbates the decline. Perhaps ‘Declines in population density could be exacerbated by culture loss, thereby linking culture to conservation.’ You make this point in this sort of wording at L.36 and L.284

L.15-18 These sentences could be reworded. Partly because this is the only place “fidelity” is mentioned in the ms and partly because this is where a clear summary of the main findings is usual. ‘Wild males at low population density tend to sing atypical songs, either unusual for the area (27%) or resembling other species’ songs (16%).’

L.18- 19 This sentence combines two themes and splits the link between atypical song production and fitness. You could move up the L.19-22 result, perhaps expressed as ‘Males singing atypical songs were less likely to pair and nest than males that sang the regional cultural norm.’

and follow with the L.18-19 sentence, split into 2 sentences.

These changes would lead very neatly into the penultimate sentence – perhaps starting ‘We therefore ...’

L.19-22 see previous comment

L.23-24 Consider strengthening the concluding sentence by adding ‘and therefore provide a useful conservation indicator.’

L.34 Perhaps rephrase part of sentence to acknowledge that there is some evidence for vocal culture degradation in species other than humans, examples include Holland et al 1996, *J. Avian Biology* 27, 47-55; Osiejuk & Ratynska 2003, *Folia Zool* 52, 275-286.

L.47 It would help many readers to have an idea in km of what constitutes long distances in this context, perhaps replace “long distances” with ‘100s km’ or ‘1000s km’

L.52-54 It is perhaps worth making the point that this is probably not unusual in the wild, as several species have been documented as learning songs from their territorial neighbours rather than fathers (there are several references in the section on Song learning in birds in Garland & McGregor 2020 *Frontiers in Psychology* doi: 10.3389/fpsyg.2020.544929), so it is likely to be a general issue in song learning in declining populations.

L.63 As at L.47 it would be help many readers to replace “vast” with a numerical indication of range e.g. 1000s km²

L.80 The part of the sentence after “and” isn’t really a sequitur (unique bands don’t sex an individual). It seems likely that individuals were sexed in the hand during banding (because individual identification is dealt with in lines 82 onwards), so this part of the sentence could be replaced with ‘and in the hand during marking with unique combinations of coloured leg bands’.

L.86 See general comment on captive birds above. Given the importance of location to this study, at this point in the ms it would be helpful to note where captive birds were released.

L.92 ff It is not clear to me from the Supp. file S2 why the second sentence cannot be combined with the first. Could “...squeak; and a song etc” be replaced with ‘... squeak; and a highly distinctive song, consisting of ... warble (Supplementary file S1) produced with characteristic head-bobbing (Supplementary file S2).’

L.97 (also 115) Please define / expand “quality” so that the reader understands what you consider to be high quality (e.g., lack of background noise, high signal to noise ratio, recorded from within ?20m etc). The reason that this is important is to allow the reader to judge whether this selection process could have biased the data used in analysis (see the STRANGE framework on sampling bias Webster & Rutz 2020 *Nature* 582, 337-).

L.101-104 See request above for more detail on this aspect of the study. Perhaps the phrase “failed to sing any species-specific songs and instead produced the song of a different bird species” could be reworded to include some of this detail, for example ‘were heard to sing songs that resembled another species. Eight of these ‘interspecific singers’ were recorded.’

L.109-110 Perhaps this sentence could be made clearer, as “obtained” may refer to recordings made by the experienced observer (where the observer’s experience is relevant) and by others (where the first part of the sentence is not relevant).

- L.117 It seems unlikely that your aim was “to reduce signal to noise ratio”, more likely you aimed ‘to increase signal to noise ratio’?
- L.118 and ff The term “sonogram” is used in this line at in a table heading (S1), whereas spectrogram is used at L.170 and in Figure 2 caption. Spectrogram is the more standard usage and would be better used throughout the ms for consistency.
- L.134 Remove “vast” as it is an unreferenced adjective and the detail will now be available after change at L.47
- L.137 Why 14 attributes here when 15 were used previously (L.127)?
- L.143 Explain how they could be assigned with high confidence – perhaps they had been clearly / routinely heard by the experienced observer? This also relates to the categorization of “unknown” song types.
- L.149 Replace “... males’ ...” with ‘... male’s ...’
- L.160 Please say how small. E.g., ‘The small (5) sample ...’
- L. 164 Perhaps expand “distance” to ‘song similarity distance’ to prevent confusion with geographic distance.
- L.170 Please expand detail. Were songs “readily recognizable” by the experienced observer? By all the authors?
- L.171-173, 175 & 179, 185 This is a key result, but difficult to understand in its current form even read alongside Fig1A and with reference back to male totals in the Methods. Please reword using numbers of males singing the variant and the number of all males in that area. I can’t work these numbers out from Figures + Methods. Something like the following would be clearer ‘In the Blue Mountains ?? of ?? males sang the typical Blue Mountains song and this song was not found elsewhere. In the Northern Tablelands ?? of ?? males, sang the Northern Tablelands song and 6 males sang it elsewhere.’
- L.180 See general comment above on song similarity as evidence of learning. An accurate and neutral reporting of the data would be ‘Throughout the study area, 17? males sang interspecific songs: 4 resembled songs of noisy friarbird, 5 little wattlebird,’ etc etc
- L.193 “significantly” implies a supporting statistical analysis, replace with ‘noticeably’
- L.197 To be clearer could replace “Repeat recordings ...” with ‘Recordings of the same individual within a year were ...’
- L.199 “remained consistent” see general comment above
- L.203 Which males are “the other 21 individuals in the same season” referring to? Are these the captive population? This detail needs to be clarified and more obviously related to the surrounding text.
- L.204 “field-validated” see general comment above
- L.224 Can combine sentences “... with a circle. The size ... corresponds” as ‘... with a circle corresponding’
- L.227 Can remove “spatial windows of <”
- L.216 and throughout captions, ensure label e.g., (1), (A) occurs before the item they refer to.
- L.232, 233 Also for subsection labels (A-E) Species-specific etc
- L.237 see general comment above on axis scaling for comparisons
- L.275 Add information on what the line with shading indicates in (C) and (F)
- L.325 Add of total interspecific (Five out of 18 of the interspecific singing ...) to aid interpretation.
- L.343 Is it known whether Hawaiian honeycreepers became fragmented and widespread (like regent honeyeaters) or concentrated into a smaller area of suitable habitats? If the latter, this could explain song convergence.
- L.353-355 Perhaps replace, or add, a comment on the practical conservation value – indications of a loss of culture may be a useful conservation tool to establish the level of threat faced by declining populations?

Author's Response to Decision Letter for (RSPB-2020-2678.R0)

See Appendix A.

RSPB-2021-0225.R0

Review form: Reviewer 2

Recommendation

Accept with minor revision (please list in comments)

Scientific importance: Is the manuscript an original and important contribution to its field?

Excellent

General interest: Is the paper of sufficient general interest?

Excellent

Quality of the paper: Is the overall quality of the paper suitable?

Excellent

Is the length of the paper justified?

Yes

Should the paper be seen by a specialist statistical reviewer?

No

Do you have any concerns about statistical analyses in this paper? If so, please specify them explicitly in your report.

No

It is a condition of publication that authors make their supporting data, code and materials available - either as supplementary material or hosted in an external repository. Please rate, if applicable, the supporting data on the following criteria.

Is it accessible?

Yes

Is it clear?

Yes

Is it adequate?

No

Do you have any ethical concerns with this paper?

No

Comments to the Author

Overall, the revisions have addressed my comments well. There are a few places in the new text where I think changes would benefit the ms. These are referred to by to the authors' numbered list, e.g. 14), or line numbers of the tracked changes proof file.

l. 96 "dated" would be better

l. 115-118 (response to 14) and 15)) A small change to the wording at two places in this part of the text would maintain the distinction between what you observed and what you have inferred (and perhaps also fulfil the editor's request to moderate claims).

- 1.115 “produced the song of as different species” is an inference, “sang songs we considered similar to a different bird species” is what you observed.
- 1.117 “the species whose songs they had learned” is an inference, “the species we considered most similar” is what you observed.

1.118 add initials in brackets to identify the experienced observer ?RC, as you have done elsewhere

1.230 check dates, it says 1986-2012 at line 96. In Fig captions 1.264 and elsewhere (e.g. 1.275 and S2 text) it says pre-2012.

1.391 (also 67) and response to 67)) It seems a shame not to enlighten the reader on a likely reason for the difference with honeyeaters. 5 extra words could do it e.g. “ ... as the population declined, possibly because of range contraction”

1.454, 531, S5 lines 1,3,7 typo: replace “signing” with “singing”

Supplementary Information

S1 para 2 line 5 change “... juvenile males there therefore isolated from ...” to “... juvenile males are therefore isolated from ...”

S1 para 2 line 10 change “...within earshot of the sounds multiple other bird species ...” to “...within earshot of the sounds of multiple other bird species ...”

Table S6, left hand column head change “Xeno-canto” to “xeno-canto”

Fig. S3 Present spectrograms with same x-axis scaling as Fig.2.

Decision letter (RSPB-2021-0225.R0)

12-Feb-2021

Dear Mr Crates

I am pleased to inform you that your manuscript RSPB-2021-0225 entitled "Loss of vocal culture has fitness costs in a critically endangered songbird" has been accepted for publication in Proceedings B.

The referee(s) have recommended publication, but also suggest some minor revisions to your manuscript. Therefore, I invite you to respond to the referee(s)' comments and revise your manuscript. Because the schedule for publication is very tight, it is a condition of publication that you submit the revised version of your manuscript within 7 days. If you do not think you will be able to meet this date please let us know.

[http://datadryad.org/submit?journalID=RSPB&manu=\(Document not available\)](http://datadryad.org/submit?journalID=RSPB&manu=(Document not available)) which will take you to your unique entry in the Dryad repository. If you have already submitted your data to dryad you can make any necessary revisions to your dataset by following the above link.

Please see <https://royalsociety.org/journals/ethics-policies/data-sharing-mining/> for more details.

Sincerely,
Dr Sasha Dall
mailto: proceedingsb@royalsociety.org

Reviewer(s)' Comments to Author:

Referee: 2

Comments to the Author(s).

Overall, the revisions have addressed my comments well. There are a few places in the new text where I think changes would benefit the ms. These are referred to by to the authors' numbered list, e.g. 14), or line numbers of the tracked changes proof file.

1. 96 "dated" would be better

1. 115-118 (response to 14) and 15)) A small change to the wording at two places in this part of the text would maintain the distinction between what you observed and what you have inferred (and perhaps also fulfil the editor's request to moderate claims).

- 1.115 "produced the song of as different species" is an inference, "sang songs we considered similar to a different bird species" is what you observed.

- 1.117 "the species whose songs they had learned" is an inference, "the species we considered most similar" is what you observed.

1.118 add initials in brackets to identify the experienced observer ?RC, as you have done elsewhere

1.230 check dates, it says 1986-2012 at line 96. In Fig captions 1.264 and elsewhere (e.g. 1.275 and S2 text) it says pre-2012.

1.391 (also 67) and response to 67)) It seems a shame not to enlighten the reader on a likely reason for the difference with honeyeaters. 5 extra words could do it e.g. "... as the population declined, possibly because of range contraction"

1.454, 531, S5 lines 1,3,7 typo: replace "signing" with "singing"

Supplementary Information

S1 para 2 line 5 change "... juvenile males there therefore isolated from ..." to "... juvenile males are therefore isolated from ..."

S1 para 2 line 10 change "...within earshot of the sounds multiple other bird species ..." to "...within earshot of the sounds of multiple other bird species ..."

Table S6, left hand column head change ""Xeno-canto to "xeno-canto"

Fig. S3 Present spectrograms with same x-axis scaling as Fig.2.

Author's Response to Decision Letter for (RSPB-2021-0225.R0)

See Appendix B.

Decision letter (RSPB-2021-0225.R1)

19-Feb-2021

Dear Mr Crates

I am pleased to inform you that your manuscript entitled "Loss of vocal culture and fitness costs in a critically endangered songbird" has been accepted for publication in Proceedings B.

Your article has been estimated as being 9 pages long. Our Production Office will be able to confirm the exact length at proof stage.

Open Access

Paper charges

Sincerely,
Editor, Proceedings B
<mailto:proceedingsb@royalsociety.org>

Appendix A

Australian
National
University

Fenner School of Environment and Society
Linnaeus Way
Canberra
Australia 2601
28th January 2021

Dear Dr Dall,

Many thanks to yourself, the associate editor and the two referees for their very detailed comments on the original version of our manuscript titled 'Loss of vocal culture has fitness costs in a critically endangered songbird.' We are pleased that you see value in the manuscript and grateful for the opportunity to submit a revised version. As such, please find attached a substantially revised manuscript, now titled 'Loss of vocal culture *and* fitness costs in a critically endangered songbird' in which we have addressed all of the comments raised by both reviewers and the associate editor.

The main points raised by the reviewers and the associate editors were:

- Further details on the methodology: we have added a 'Supplementary Methods' document which provides additional information on the captive breeding programme, historical song recordings, the recordings of other species songs, production of the spectrograms and further explanation of the samples included in each analysis.
- Moderation of claims of causation: in addition to the subtle title change, we have removed terms such as 'linked to' and replaced with terms such as 'associated with' throughout the manuscript. Moreover, we address this issue directly in a paragraph towards the end of the discussion.
- General writing clarity: we have incorporated many of the rewordings suggested by reviewer two, and have attempted to improve clarity of writing throughout the document.
- Data accessibility: we have been uploaded all sound files, datasets and annotated code to Data Dryad. We have also emailed the Macaulay library regarding uploading larger sound files, and are waiting to hear back from them regarding their sensitive species data policy.

Please find below the comments from the associate editor and the two reviewers (numbered and italicised). Under each numbered reviewer comment we state in bold how we have addressed the comment in the revised manuscript. Line reference numbers refer to the version of the revised manuscript with track changes.

We hope the major changes we have made are sufficient for you to consider our manuscript now worthy of publication in *Proceedings B*.

Yours sincerely,

Dr Ross Crates (on behalf of the co-authors). www.difficultbirds.com

Associate Editor

Board Member: 1

Comments to Author

Both reviewers feel that this is an important study, but that there is substantial room for improvement, particularly in the way that the data is presented. Also some claims are made that could be moderated somewhat, and their importance made more accessible.

One reviewer feels that some of the methods used require more information and more explanation - these could be provided at length in the SI, as well as explained more clearly in the main text.

This reviewer also raises important issues in interpretation (esp in terms of song classification, and the analysis of historical as well as contemporary songs together, and in potentially overstating your conclusions (particularly in terms of inferring cause from associations) that you should address in any new version.

Reviewer(s)' Comments to Author:

Responses to reviewer comments are provided in bold under each respective comment below.

Referee: 1

Comments to the Author(s):

Review of Crates et al. Loss of vocal culture has fitness costs in a critically endangered songbird

This ambitious manuscript describes the results of research attempting to link population declines of an endangered songbird with cultural loss of sexual signals and then further show that this cultural loss in turn has fitness consequences. The authors document a number of male birds that sing atypical songs and show that these males tend to be more isolated from other birds and less likely to be paired or have nests, suggesting a relationship between these three variables. I think this subject is of broad interest and I found the manuscript to be generally well-written. Unfortunately, I think there are some issues with the manuscript as well.

The first issue is that the authors overstretch the interpretation of their data. Ultimately, the authors have demonstrated correlations between three traits (population size, song type, and fitness), yet the manuscript implies causation (reduction in population size \rightarrow poor learning opportunities \rightarrow reduced fitness). This is certainly one logical interpretation, but it very well could be the case that the causal relationship goes in a different direction, is more complex, or has a different mechanism altogether. To me, terms like "linked to" imply causation, which does not seem appropriate here.

Second, while the manuscript is well written, it is often missing important information, which sometimes makes it difficult to evaluate the details of the methods. One really great thing about this paper is that it is really ambitious in terms of how many topics and types of data it

synthesizes. However I think that this breadth also makes it challenging to present all of the relevant details of all parts of the study, and this is especially true in a relatively short form like Proceedings. I think more clarity is needed in many of the experimental details, for example how songs were evaluated, whether birds were banded, which birds were included in different samples, etc. (more detailed comments below by line number).

Last, I had a few concerns about the methodological approach and experimental logic. As I mentioned above, details are not always clear in the manuscript, so I'm not entirely sure if I am always interpreting the methods correctly, but I think that a couple of the issues listed below could be quite important, especially those dealing with how song types are classified, which birds are included, and whether it is appropriate to compare the historical songs with contemporary songs because of the date range of historical songs and the geographic range of contemporary songs.

Below I provide more details about these concerns, and other more minor issues, by line number:

1) L19 (and others): The authors do well to explicitly point out that correlation does not imply causation in the discussion section, but the rest of the manuscript is written as though there is a causal link between the variables. Terms like "linked to" imply causation, but this cannot actually be inferred from the data. I think the authors need to tone down their wording so as not to imply causation between correlated variables.

1) Here and throughout the revised manuscript we have replaced terms such as 'linked to' with 'associated with'. In the title we have reworded 'has fitness costs' to the more neutral 'and fitness costs'. (See also lines 20, 329, 343 and 399). We also address this issue directly in the discussion (Line 378).

2) L34. While not commonly studied, there are at least a few examples of avian vocal culture changing in small populations. A few studies that come to mind are below, though there are probably others.

Laiolo, P., Vögeli, M., Serrano, D. and Tella, J.L., 2008. Song diversity predicts the viability of fragmented bird populations. PLoS One, 3(3), p.e1822.

Ortega, Y.K., Benson, A. and Greene, E., 2014. Invasive plant erodes local song diversity in a migratory passerine. Ecology, 95(2), pp.458-465.

(I note that this one is cited in the discussion). Valderrama, S.V., Molles, L.E. and Waas, J.R., 2013. Effects of population size on singing behavior of a rare duetting songbird. Conservation Biology, 27(1), pp.210-218.

*Martínez, T.M. and Logue, D.M., 2020. Conservation practices and the formation of vocal dialects in the endangered Puerto Rican parrot, *Amazona vittata*. Animal Behaviour, 166, pp.261-271.*

2) We have added the Laiolo et al. 2008, Ortega et al. 2014 and Valderrama et al. 2013 studies to the citation list here and reordered the citations accordingly (Line 40).

3) 86: perhaps a bit more info about the breeding program would be useful here. For example, how many birds are released, and what proportion of the population is this? What are the rearing conditions and song learning opportunities in captivity? Etc.

3) We have added a reference here to a new Supplementary Methods section, which provides more details on the captive breeding facilities (Supplemental Text S1 and Supplemental Table S5).

4) 77/80/95: are all birds included in the database uniquely marked? This is not reported, but it seems important to ensure that all of the birds included in the study are unique individuals and, given that this species is nomadic, this seems difficult to be sure of unless all birds are banded.

4) All captive-reared males are uniquely marked and we made every effort to mark as many wild males as possible. Whilst we agree with the reviewers' point that it would be desirable to mark every wild male, given the nomadic nature of the study species, the large distances between sightings locations and the fact that most birds are detected during the breeding season, it is not logistically or ethically possible to safely capture and mark every wild male regent honeyeater. The information we provide in the methods section regarding song repeatability shows that the proportion of marked birds resighted across years during the study is low (approximately 5%). Thus, whilst it is possible that some unmarked wild males were recorded twice in different seasons, we are confident that the proportion of these males is small, not spatially biased and therefore very unlikely to have a substantial effect on the results of our study. We have added in parentheses the number of males recorded that could be identified via colour bands. (Line 91).

5) 96: I'm assuming that a given male of this species sings only a single song type, rather than a repertoire. Is this correct? Please state this explicitly and provide a citation.

5) Yes, a given male does sing only a single song type. However given this is the first standardised study of wild regent honeyeater song culture, there is no citation available that states as much. Hence, we reported in the 'song repeatability' section of the results in the original submission that "Individual regent honeyeaters' songs remained consistent and faithful to a single song type over time." (Previously line 212) and now reworded in the revised version as "Individual regent honeyeaters' consistently produced only one song type over time." (Line 236).

6) 110: please clarify that these 7 songs were not unequally distributed across the sampling populations.

6) The 7 song types were unequally distributed across the sampling populations because they were originally based on either spatial location (Typical and Clipped Blue Mountains, Northern Tablelands and Captive-bred) or time period (pre-2012). We have expanded the song classification section to help clarify this (Line 101) and then describe

more clearly the distribution of song types in the results section: “In the Blue Mountains, 93 of 132 males sang the typical Blue Mountains song and this song type was not found elsewhere. In the Northern Tablelands, 17 of 22 males sang the Northern Tablelands song and 6 males sang this song type in the Blue Mountains, having likely dispersed there (figure 1A).

Some males produced song types that were atypical for their region (figures 1A & 2): Located exclusively in the Blue Mountains, 20 males produced a distinctive, abbreviated version of the typical Blue Mountains song (figures 1A & 2D). We therefore classified these birds’ songs as their own song type- the ‘clipped Blue Mountains’ song. Located throughout the study area, eighteen males sang interspecific songs: five males’ songs resembled songs of little wattlebird *Anthochaera chrysoptera*, four of noisy friarbird *Philemon corniculatus*, three of spiny-cheeked honeyeater *Acanthagenys rufogularis*, two of pied currawong *Streptera graculina*, and singles of eastern rosella *Platycercus eximius*, little friarbird *Philemon citreogularis*, olive-backed oriole *Oriolus sagittatus* and black-faced cuckooshrike *Coracina novaehollandiae* (figures 1A & 2F-O).” (Line 201).

7) 95-115: *I’m a bit confused about the sample sizes reported here. Early in this section it is reported that out of 251 males, there were 161 males who sang yielding 47 quality recordings, but later on it is reported that there are 73 wav files that were of high quality to analyse. I’m especially confused about the 47 vs 73 discrepancy. I’m also confused about the additional birds who were studied but not recorded—were assessments made about these based simply on how they sounded to the observer or were these birds excluded from the analyses? The former seems quite problematic, but if the later I don’t really understand why they are included in the manuscript.*

Having now looked at the figures, I’m coming back to this comment as I would think it would be quite difficult to discern some of these song types from ear alone—some types are quite similar to the interspecific songs, for example. It is possible that the authors can do this, but I think including some more information about how the authors have ensured that these field assessments (if used) are reliable and repeatable would be important to include.

7) We understand the reviewers concern, as it was challenging to explain succinctly which individuals and which recordings could be included in each analysis. We have reworded (and corrected where necessary) the sample sizes stated. The reviewer’s initial confusion regarding the discrepancy between the sample sizes of 47 and 73 was because the 73 recordings in the spectral analysis also included the 12 captive birds, plus repeat recordings of some individuals (both wild and captive) for song repeatability assessment. To eliminate any further confusion we have uploaded to Dryad a summary excel file with each individual male in rows (with some duplicates for males located in different years), and columns describing metadata such as location, song type, colour band combination, sound file code etc., as well as additional columns detailing how each male could be identified and whether he could be included in each of the statistical analyses we conducted. We also now provide reference to this table at the end of the methods section: “See data availability section below for access to metadata detailing how we identified individuals and which individuals we included in each component of the statistical analysis.” (Line 191).

With experience, song types are much easier to discern than the reviewer may have feared through visualising the spectrograms in Figure 2. We can confirm that the songs of the interspecific singers shown in Figure 2 sounded nothing like any of the species-specific songs we recorded in the wild or in captivity over the past five years. To reinforce that our classifications of interspecific singers were not subjective, we included stratified, random samples of the interspecific songs in a blind song classification test of six professional ornithologists who were able to not only identify interspecific songs with 89% agreement with our own classifications, but were also able to correctly identify 79% of the model species whose songs the interspecific singers most closely resembled (using only a single one to two second song recording and no background information or field context). See Supplemental Text S4 and Supplemental Table S2, including reference to the full classification test dataset in Dryad.

Having now uploaded the sound files to Dryad, we would also encourage anyone with concerns regarding song classification to listen to the recordings as we feel this will help allay any remaining concerns reviewers or readers may have.

8) 144: *why could these males' songs not be classified to a song type?*

8) Importantly, the reason these males' songs were unknown is not because their songs were intermediate between different song types, but because we did not hear them sing. For example, a small number of males were already nesting at the time they were detected and males cease singing when the first egg is laid. For others, we may have had a flock including multiple males, but could only confidently assign the song type to some of them. So for example, we may have identified six males in a flock at a location, assigned a song type to three males and left the other three birds' song types as 'unknown'. We have expanded the following sentence to clarify: "We classified the songs of a further 63 males, whose songs we could not assign to a song type as 'unknown' because we did not hear or record these males singing at the time they were detected, and not because their songs were 'intermediate' between song types." (Line 171, see also Supplemental Texts S3 and S6).

9) 241/Fig 3: *The authors test the idea that songs have become less complex over time by examining the pre 2012 songs with the current songs. One concern is that there are only 14 songs recorded in the entire period of 1986-2011. That seems like a pretty small sample for judging historical patterns. Also, if there are statistical changes in the songs from 2012-2018 (6 years), then is it really appropriate to lump together songs from 1986-2011 (25 years) and assume that they have not changed during this time? No information is provided, that I can find, about when in this period these songs were recorded, but given the changes proposed in more recent songs, this seems like critical information to include.*

9) We agree with the reviewer that it would be desirable to have a larger sample size of historical recordings, however these birds were recorded opportunistically, hence 14 is the maximum sample of high quality recordings we were able to accrue. Given the limited sample size, we did not attempt to make any more detailed inferences about historic song other than that they demonstrate that wild birds' songs were historically more complex prior to 2012 than the songs of the entire remaining contemporary wild (and captive) population. (See Supplemental Text S2 and 'Recordings_summary.xlsx')

file uploaded to Dryad.

10) Next, I'm not really clear about the expected findings for this analysis given that all of the historical songs were recorded in one population in the Blue Mountains. Above caveats aside, I'm not sure this analysis is an appropriate way to test the question about whether songs have become less complex over time. I can see how comparing historical Blue Mountain songs with contemporary Blue Mountain songs would be interesting and address this question, but why should there be a relationship between historical Blue Mountain songs and contemporary songs from other populations? Is there a reason to think that all birds used to sing the Blue Mountain dialect no matter where they lived? Otherwise, there are too many variables changing between these samples and while it is fine to note that some dialects/populations are more complex than others it is not appropriate to then equate this with a loss in complexity over time as implied.

10) We tried to minimise the spatial bias in the historical recordings by only including birds recorded in the Blue Mountains. We appreciate the reviewers concerns regarding comparisons of these recordings to contemporary song types, however we feel that the comparisons are valid between the historical recordings and at least three of the four contemporary, species-specific song types (i.e. typical Blue Mountains, clipped Blue Mountains and captive-bred) because (i) we recorded these contemporary wild birds in the same region as the historic recordings or (ii) the captive population was established and supplemented with birds from the Blue Mountains. We have also corrected the x-axis label in Figure 3 from 'Pre-2012' to 'Pre-2012 Blue Mountains.' (Line 294).

11) 295: The authors emphasize the importance of a critical learning period here and at other places in the manuscript. Is there evidence that this species has a critical period or when this period takes place?

11) We have a PhD student currently studying song learning in the captive regent honeyeater population. However to date there are no peer-reviewed publications available to cite, so our assertions are based on (i) the preliminary unpublished findings of the PhD project; (ii) our own results whereby we found no individually marked adult males that changed their song type over time; and (iii) the broader literature (e.g. Beecher and Brenowitz 2005, Eens et al. 1992; Mennill et al. 2018).

12) 333: Interesting that the winter grounds of the birds are unknown. I wonder if birds could be associating with other individuals at this time, potentially providing opportunities for mate acquisition outside the breeding season or even song learning? As the authors note, the song learning must take place after dispersal from the natal territory, so could some of the song learning take place at these, unknown, locations?

12) Yes, the unknown wintering grounds exemplify the challenges associated with studying regent honeyeaters! Given our knowledge that regent honeyeaters disperse from the breeding grounds in summer when juveniles are typically between 2 and 4 month of age, it is likely that a major proportion of song-learning occurs on the non-breeding grounds. This is supported by the fact that in spring 2020 we have observed a regent honeyeater in the Northern Tablelands breeding area (near Glen Innes,

approximately 180km inland) singing the song of a little wattlebird which is almost exclusively a coastal species. Thus, we assume this individual spent the winter on the coast, associated with and learned the song of a little wattlebird, then subsequently dispersed/returned to the Northern Tablelands breeding grounds. eBird records indicate there are no contemporary records of little wattlebirds within 70km, and less than 20 within 100km, of where we detected the regent honeyeater singing this species' song. Because of the anecdotal nature of such observations, our capacity to discuss them in the manuscript is limited, however we have added this comment and our response as a text in the Supplemental Information (Text S7) and cited it in the discussion (Line 343).

Referee: 2

Comments to the Author(s)

This study documents the song variation shown by male regent honeyeaters, a critically endangered songbird, in relation to their population density – at low densities, males are more likely to sing atypical songs (either not the commonest song of the area, or a song resembling that of another species), and such males are less likely to pair and build a nest. In addition, songs of captive-raised males are even more atypical. Given that song is important for reproduction in all songbirds, these results likely have conservation consequences for the study species. The study also has important wider implications for a least two large areas of interest. First, the UNEP has recognized the potential impact of culture in conservation and called for more evidence – this study is a very important example of such impact and as such is likely to be widely cited in the conservation literature across all taxa. Second, as a relatively rare example of song acquisition in the wild this study adds important information relevant to song learning. Lab studies predominate in the song learning literature and such studies invariably design out social interaction and focus on early life. Yet studies in the wild generally point to the importance of social interaction and have shown learning in most species extends well beyond the nestling phase. This study's well-documented examples of relatively common singing of other species' songs is particularly welcome, since most other information on this behaviour is usually in the form of a short note with little or no supporting acoustic information (usually the birds were thought to sound similar, but were not recorded).

In summary, this is an important piece of research that is likely to be commonly cited in a number of areas of current research and it has practical conservation implications. However, the clarity of writing and the level of detail can be increased, and this should help ensure that the study is as widely read and cited as it deserves to be.

General comments

13) Interspecific and captive songs. Your study is unique in combining such songs with information on songs of contemporary singers in the wild and, as you point out around L.299-302, reports of interspecific singing are usually single individuals, so the level you report in honeyeaters is unprecedented. To make the most of the insights these males' songs

could provide needs some more detail to be added to the text.

For the captive facility the detail should include any factor that could influence song development (location, housing conditions, in flocks, alone but in earshot, presence of other species etc) and the source of original captive breeders etc.

13) We have added this additional information on the captive breeding program as a section in the new Supplemental Methods document and provide a reference to the document as follows: “We recorded captive-bred birds either shortly after their release into the wild in 2017 or in captivity in August 2019. See Supplemental Text S1 for further details of the captive breeding program.” (Line 98).

14) For interspecific song, detail on the following would allow the reader to better assess the reported similarities at a number of places in the text:

• in section (c) of Methods report where the songs of other species used for comparison came from (the memory of an experienced observer is an OK answer)

We have expanded the sentence to say “We classified these birds as ‘interspecific singers,’ based on either visual similarities between spectrograms of the songs of interspecific singers and of the species whose songs they had learned (n = 8) or an experienced observer’s knowledge of the vocalisations of the local avifauna (n = 10).” (Line 118). We have also added a section to the Supplemental Information- texts S4 to S6 and Table S6 that describes further the classification of interspecific songs and provides details on the location and time of the reference songs we obtained from xeno-canto to produce figure 2.

15) • L.180 in Results you should report the result you found (assessed visual / acoustic similarity of spectrogram / heard in field respectively between study species and another species) rather than an interpretation (“that had learned the songs of”).

We have reworded and expanded the sentence as follows: “Located throughout the study area, eighteen males sang interspecific songs: five males’ songs resembled songs of little wattlebird *Anthochaera chrysoptera*, four of noisy friarbird...” (Line 211). We also state now in the methods section that “We classified these birds as ‘interspecific singers,’ based either on visual similarities between spectrograms of the songs of interspecific singers and of the species whose songs they had learned (n = 8) or knowledge of the songs of the local avifauna in an experienced observer (n = 10).” (Line 118).

16) • L.216 As previous comment, a form of words such as ‘Images of the other species with song most similar to honeyeater ...’

16) We have reworded this sentence in the legend to Figure 1A as follows: “The species whose songs each interspecific singing regent honeyeater most closely resembled are shown:.....” (Line 256).

17) • L.235-6 *Add location of xeno-canto song, or perhaps how far from the particular male honeyeater. As your study and many others have shown how variable even “species typical” songs are, please state how these songs were chosen. Did they particularly look like / sound like that specific honeyeater song? This level of detail is important to allow the reader to assess the significance of the similarity you are drawing attention to.*

17) We have added this further detail on the xeno-canto recordings to the new Supplemental Material Table S6, which is now cited in the figure 2 legend “ See Supplemental Information text S5 & S6 for further information on other species’ songs and spectrograms, respectively.” (Line 282).

18) • L.296 *“other species that they may happen to associate with” suggests that you have data on the presence of the other species with honeyeaters and it would help interpretation to know what that is, with direct field observations giving more weight to the interpretation than an overlapping geographical range known from the literature.*

18) Because regent honeyeaters disperse from the breeding grounds to largely unknown areas, we do not have any information or direct observations of interspecific singing regent honeyeaters associating with (or learning songs from) individuals of the species whose songs they most closely resemble. All the other species are relatively common, with ranges that overlap to a large extent with that of the regent honeyeater. See our response to comment 12 above and Supplemental Text S7 for further information.

19) *Qualitative v. quantitative measures of song similarity. These differ considerably in the level of detailed methodology you report; with access to the song measures data set and the detail on quantitative methods used, it would be possible to replicate your analyses. Two aspects of the qualitative analyses mean that replication would not be possible. The first is the data set, given the issue you have had with uploading zip files, could you consider depositing the recordings you analysed in an online archive (best if maintained by one of the big sound archives like Cornell).*

19) We have uploaded the sound files all the data and the R script to Dryad. We have edited the data accessibility section accordingly: “Data, sound files and an annotated R-script are available via the Dryad Digital Repository (<https://doi.org/10.5061/dryad.mkkwh70zj>).” (Line 408).

We have also emailed the Macaulay Library to enquire about the possibility of uploading longer song recordings without publicly disclosing location information. We are awaiting response from them and will hopefully be able to upload recordings in the near future.

20) *The second is that there is no comparable detail on the qualitative analysis to the R version, package (in ms) and scripts (in Supplementary data).*

20) We state in the methods that “A single observer with 6 years’ experience of monitoring regent honeyeaters (RC) obtained all but seven contemporary song

recordings.” (Line 124) and then go on to state “we included in the dataset males whose songs were either not recorded or were not recorded of sufficient quality for acoustic analysis, but could be assigned with high confidence to a song type in the field (n = 105) because they were clearly heard singing by an experienced observer (RC).”

To provide further evidence that our song classifications and recognition of the species whose songs interspecific singing regent honeyeaters most closely resembled, we conducted a blind song classification test on six professional ornithologists with varying degrees of experience with wild and captive regent honeyeaters. These people were able to assign 20 of the song recordings to the same song type with an average 89% agreement with our own classifications. Details of the blind song classification task are provided in Supplemental Text S4, Table S2 and in an additional paragraph in the song classification section of the methods: “To quantify the repeatability of our song classifications, we asked six professional ornithologists to assign blind a stratified, random sample of 20 songs to the contemporary song types and calculated the percentage agreement between our classification of each song and the classifications provided by the participants. We also asked each participant to identify the model species, if they thought that a recording was of an interspecific singer, and calculated the percentage agreement between our identification of the model species and that of the participants. See Supplemental Text S4 for further information on the blind song classification procedure.” (Line 126).

21) Relevant detail could be added throughout the ms (or as Supplementary material if space is too limited). Two specific points in the text where detail is needed are noted below (L.143, L.170). Simply expanding terms like “remained consistent” at L.199 and “field-validated” etc L.204 would be insightful.

21) With regard to L143 (defining ‘sufficient quality’), we have added the following sentence at the point we first mentioned recording quality (L96 in the original submission): “.....able to obtain quality recordings, defined as a high signal to noise ratio and no other background noises (so that all elements of the song were clearly visible in the spectrograms), of the songs of 47 of them.” See also our response to comment 42 below.

With regard to L170, we have reworded the sentence as follows: “These two song types were also readily audibly recognisable by an experienced observer or through visual inspection of spectrograms.” (Line 197). We have added a paragraph in the song classification section of the methods describing the song classification task: ““To quantify the repeatability of our song classifications, we asked six professional ornithologists to assign blind a stratified, random sample of 20 songs to the contemporary song types and calculated the percentage agreement between our classification of each song and the classifications provided by the participants. We also asked each participant to identify the model species, if they thought that a recording was of an interspecific singer, and calculated the percentage agreement between our identification of the model species and that of the participants. See Supplemental Text S4 for further information on the blind song classification procedure.” (Line 126) and the results of this task: “Using only a single song recording and with no field context (i.e. without any capacity to observe birds singing in the wild or in captivity), there was 89% agreement between our classification of song types and the classifications assigned by six professional ornithologists. For interspecific singing regent honeyeaters, the

participants identified the same model species as us in 79% of cases (Supplemental Table S2).” (Line 218).

See also our response to comment 52 below.

We have also made substantial changes to the wording of the ‘song repeatability’ section of the results (c/f L199 and 204). This section now reads as follows (see also responses to comments 56 & 57 below: “Individual regent honeyeaters’ consistently produced only one song type over time. Repeat recordings of the same individual’s song were more similar to each other than to those of all other individuals (Mantel test, $n = 25$, $Obs = 0.028$, simulated $p = 0.015$). We recorded two colour-marked males in different years; one male produced the typical Blue Mountains song type in 2015 and 2017, and another produced the clipped Blue Mountains song type in 2016 and 2017 (figure S3). Two males first recorded in the wild producing a typical Blue Mountains song in 2019 maintained this song type in captivity at least 18 months later, having been recruited to the captive population. We obtained repeat recordings of the songs of 21 individuals in the same season, all of which sang the same song type over time. We also observed a further five colour-marked males across years, whose songs we could not record but could consistently assign by ear to the typical Blue Mountains song type.” (Line 236).

22) “Tutor”. I strongly recommend replacing this term in captions for Figures and Tables, the third column of Fig.2, and at most places in the text (the exception would be when reporting lab learning experiments that have used the term). The reason is that you are reporting similarities between songs, either seen on spectrograms or heard, and this is very different from a lab song learning experiment in which the learner is presented with a singing tutor male. Even in the lab learning case it could be argued that the term is inappropriate but using it for your results risks obscuring or confusing those results. The main issue is that the term describes only one way in which the similarity between songs could have arisen. Males A and B could sing similar songs (to our perceptions) for other reasons than A learned from B, including B learning from A, both learning from C and chance. In lab experiments it is usually possible to exclude alternatives, but in field studies this is rarely the case. It doesn’t make your results any less interesting or important to report them as similarities, assessed either qualitatively or quantitatively and it does allow the full range of possible explanations for the similarities to be considered and discussed.

22) We have replaced the word ‘tutor’ where tutor refers to interspecific song learning. In the figure 1 legend we now state “The species whose songs each interspecific singing regent honeyeater most closely resembled are shown:” and in the figure 2 legend we now state “Spectrograms of regent honeyeater song types and the songs of other species the songs of interspecific singing regent honeyeaters most closely resembled.” We agree with the reviewer that it is not an appropriate term in this respect, even if the most likely scenario is that interspecific singers are indeed using individuals of another species as a direct ‘tutor.’ However we have opted to keep the term when referring to more general, species-specific song learning. For example: “Captive juveniles are typically crèched away from adults after fledging, meaning they do not associate with

adult tutors during song learning⁶.” In this respect, tutor is a widely-used term in the song-learning literature e.g. Mennill et al. 2018, Akçay et al. 2017.

23) *Spectrograms: Settings, axis scaling and labelling. This detail is important to allow any reader to easily make the comparisons / note the similarities highlighted by the authors.*

• *The setting used by the software to produce the spectrograms is important detail to include in the Methods (sampling rate, window setting etc) because it affects both the visual appearance of the spectrogram and may affect quantitative analysis (especially if measurements are taken directly from the spectrogram).*

23) We have added this information to the new Supplemental Information text S6, with reference to it in the legend to figure 2. Please note the spectral data was not obtained from the spectrograms presented in figure 2. The spectral data was obtained from each recording through the automated *specan* function in the *warbleR* package. Our reasoning for presenting the spectrograms in figure 2 is purely to show the reader 1) the high degree of interspecific variation in song types in regent honeyeaters and 2) the acoustic similarities between the songs of interspecific singing regent honeyeaters and the corresponding songs of the species the interspecific singers have apparently mistakenly learnt.

24) • *Similarly, the axes must allow direct comparison by having the same scaling. The time axis of Figure 2 and Fig S3 vary within the figures and should be standardized.*

24) We have standardised the axes in the spectrograms in figure 2 to 2.5 seconds and a frequency range from 500 to 5500 Hz. We have standardised the repeat recordings in Figure S3 to 1.2 and 0.8 seconds, respectively.

25) • *The units of the time axis in Fig.2 should be changed to ‘Time (s)’ from “Time (s or ms)”. The same time unit should apply to all parts of Fig.S3.*

25) We have changed all the units of time in figure 2 to seconds.

26) *Frequency range and song complexity. Can you reference a study that has used frequency range as an indication of song complexity (e.g. L. 244) or provide a short explanation?*

26) We predicted that a narrower frequency range would be indicative of a decline in song complexity, however we are not aware of a citation to support this prediction so we have removed frequency range as a measure of song complexity from the revised manuscript (see edits to Supplemental Tables S1, S3 and figure S4).

Specific comments (most are suggestions to increase clarity and reduce word count to allow inclusion of necessary extra detail):

27) *Title. Consider replacing “has” with the more neutral ‘and’.*

27) Replaced as suggested. (Line 1).

28) L.12 *This sentence could perhaps be reworded. The obvious conservation issue is population density decline, the question is whether cultural loss increases or otherwise exacerbates the decline. Perhaps ‘Declines in population density could be exacerbated by culture loss, thereby linking culture to conservation.’ You make this point in this sort of wording at L.36 and L.284.*

28) Reworded as suggested. (Line 12).

29) L.15-18 *These sentences could be reworded. Partly because this is the only place “fidelity” is mentioned in the ms and partly because this is where a clear summary of the main findings is usual. ‘Wild males at low population density tend to sing atypical songs, either unusual for the area (27%) or resembling other species’ songs (16%).’*

29) We have removed the word ‘fidelity’ and reworded the sentences as follows: “Song production in remaining wild males varied dramatically, with 27 % singing songs that differed from the regional cultural norm. 12% of males, occurring in areas of particularly low population density, completely failed to sing any species-specific songs and instead sang other species’ songs.” (Line 18).

30) L.18- 19 *This sentence combines two themes and splits the link between atypical song production and fitness. You could move up the L.19-22 result, perhaps expressed as ‘Males singing atypical songs were less likely to pair and nest than males that sang the regional cultural norm.’ and follow with the L.18-19 sentence, split into 2 sentences.*

30) We have restructured this part of the Abstract as suggested: “Males singing atypical songs were less likely to pair and nest than males that sang the regional cultural norm. Songs of captive-bred birds differed from those of all wild birds. The complexity of regent honeyeater songs has also declined over recent decades.” (Line 21).

31) *These changes would lead very neatly into the penultimate sentence – perhaps starting ‘We therefore ...’*

31) ‘Therefore’ added as suggested. (Line 25).

32) L.19-22 *see previous comment.*

32) Changed as suggested. (Line 20).

33) L.23-24 *Consider strengthening the concluding sentence by adding ‘and therefore provide a useful conservation indicator.’*

33) Added as suggested. (Line 29)

34) L.34 *Perhaps rephrase part of sentence to acknowledge that there is some evidence for*

vocal culture degradation in species other than humans, examples include Holland et al 1996, J. Avian Biology 27, 47-55; Osiejuk & Ratynska 2003, Folia Zool 52, 275-286.

34) We have replaced the word ‘not’ with ‘limited’ and added the citations recommended by reviewer 1 (see our response to comment 2). (Line 39).

35) L.47 It would help many readers to have an idea in km of what constitutes long distances in this context, perhaps replace “long distances” with ‘100s km’ or ‘1000s km’.

35) We have added ‘100s km’ in parentheses after ‘long distances’. (Line 52).

*36) L.52-54 It is perhaps worth making the point that this is probably not unusual in the wild, as several species have been documented as learning songs from their territorial neighbours rather than fathers (there are several references in the section on Song learning in birds in Garland & McGregor 2020 *Frontiers in Psychology* doi: 10.3389/fpsyg.2020.544929), so it is likely to be a general issue in song learning in declining populations.*

36) We have added at the beginning of this sentence ‘As for many songbird species.....’ and included the recommended citation. (Line 57).

37) L.63 As at L.47 it would be help many readers to replace “vast” with a numerical indication of range e.g. 1000s km².

37) We stated in line 49 that the species’ contemporary range is approximately 300,000km² so we have decided to delete the word ‘vast’ instead of re-stating the range size here.’ (Line 68).

38) L.80 The part of the sentence after “and” isn’t really a sequitur (unique bands don’t sex an individual). It seems likely that individuals were sexed in the hand during banding (because individual identification is dealt with in lines 82 onwards), so this part of the sentence could be replaced with ‘and in the hand during marking with unique combinations of coloured leg bands’.

38) Changed as suggested: “Regent honeyeaters can be sexed in the field based on a combination of their size, plumage traits, behaviour, vocal attributes and in the hand (via differences in wing length and body mass) during marking with unique combinations of coloured leg bands²⁵” (Line 84).

39) L.86 See general comment on captive birds above. Given the importance of location to this study, at this point in the ms it would be helpful to note where captive birds were released.

39) We have added further details on the captive breeding program to the new ‘Supplemental Information’ file and added here: “See Supplemental Information Text S1 and S2 for further details of the captive breeding program and the historical song recordings, respectively.” (Line 97).

40) L.92 *It is not clear to me from the Supp. file S2 why the second sentence cannot be combined with the first. Could “...squeak; and a song etc” be replaced with ‘... squeak; and a highly distinctive song, consisting of ... warble (Supplementary file S1) produced with characteristic head-bobbing (Supplementary file S2).’*

40) Sentences combined as suggested. (Line 102).

41) L.97 (also 115) *Please define / expand “quality” so that the reader understands what you consider to be high quality (e.g., lack of background noise, high signal to noise ratio, recorded from within ?20m etc). The reason that this is important is to allow the reader to judge whether this selection process could have biased the data used in analysis (see the STRANGE framework on sampling bias Webster & Rutz 2020 Nature 582, 337-).*

42) We have expanded the sentence to define recording quality: “We classified the songs of 146 of these males and were able to obtain quality recordings, defined as a high signal to noise ratio and no other background noises (so that all elements of the song were clearly visible in the spectrograms), of the songs of 47 of them.” We can also confirm there is no spatial or temporal bias in the quality of the available recordings i.e. we filtered song recordings on quality blind of any spatial or temporal metadata. (Line 107).

42) L.101-104 *See request above for more detail on this aspect of the study. Perhaps the phrase “failed to sing any species-specific songs and instead produced the song of a different bird species” could be reworded to include some of this detail, for example ‘were heard to sing songs that resembled another species. Eight of these ‘interspecific singers’ were recorded.’*

42) We have expanded, in response to comment 14 above as follows: “Eighteen of the 146 males, located throughout the contemporary range, failed to sing any species-specific songs and instead produced the song of a different bird species (Figures 1A and 2). We classified these birds as ‘interspecific singers,’ based on either visual similarities between spectrograms of the songs of interspecific singers and of the species whose songs they had learned (n = 8) or knowledge of the songs of the local avifauna in an experienced observer (n = 10). We obtained quality song recordings from eight of these males.” We are reluctant to use the suggested wording as the rest of the methods is written in the first person participle.

43) L.109-110 *Perhaps this sentence could be made clearer, as “obtained” may refer to recordings made by the experienced observer (where the observer’s experience is relevant) and by others (where the first part of the sentence is not relevant).*

43) We have reworded the sentence as “A single observer with 6 years’ experience of monitoring regent honeyeaters (RC) recorded the songs of all but seven contemporary birds.” (Line 124).

44) L.117 *It seems unlikely that your aim was “to reduce signal to noise ratio”, more likely you aimed ‘to increase signal to noise ratio’?*

44) Thank you for spotting this error. We have corrected from ‘reduce’ to ‘increase.’ (Line 140).

45) L.118 and ff *The term “sonogram” is used in this line at in a table heading (S1), whereas spectrogram is used at L.170 and in Figure 2 caption. Spectrogram is the more standard usage and would be better used throughout the ms for consistency.*

45) Sonogram replaced with spectrogram throughout. (Line 141).

46) L.134 *Remove “vast” as it is an unreferenced adjective and the detail will now be available after change at L.47.*

46) Removed ‘vast’ and added ‘size’ after ‘range.’ See also response to comment 37. (Line 158).

47) L.137 *Why 14 attributes here when 15 were used previously (L.127)?*

47) This is because we did not explicitly consider one of the attribute used in the DFA to represent a measure of song complexity. N.B there are now 13 attributes as we removed frequency range in response to comment 26.

48) L.143 *Explain how they could be assigned with high confidence – perhaps they had been clearly / routinely heard by the experienced observer? This also relates to the categorization of “unknown” song types.*

48) See our responses to comments 8, 14 and 15 above. Yes the songs produced by regent honeyeaters were readily recognisable because they were clearly and routinely heard by an experienced observer (RC). We have added to the end of this sentence’ because they were clearly heard singing by an experienced observer (RC).’

See also new sections in the methods (Line 126), results (Line 218) and Supplemental Material (Text S6 and Table S6) explaining the repeatability of song type classifications based on a blind trial of 10 experienced ornithologists.

49) L.149 *Replace “... males’ ...” with ‘... male’s ...’*

49) Replaced as suggested. (Line 175).

50) L.160 *Please say how small. E.g., ‘The small (5) sample ...’*

50) n = 7 added here. (Line 187).

51) L. 164 Perhaps expand “distance” to ‘song similarity distance’ to prevent confusion with geographic distance.

51) ‘Song similarity’ added as suggested. (Line 190).

52) L.170 Please expand detail. Were songs “readily recognizable” by the experienced observer? By all the authors?

52) We expand and reword to say ‘Readily audibly recognisable by experienced observers (Supplemental Table S6).’ (Line 198).

53) L.171-173, 175 & 179, 185 This is a key result, but difficult to understand in its current form even read alongside Fig1A and with reference back to male totals in the Methods. Please reword using numbers of males singing the variant and the number of all males in that area. I can’t work these numbers out from Figures + Methods. Something like the following would be clearer ‘In the Blue Mountains ?? of ?? males sang the typical Blue Mountains song and this song was not found elsewhere. In the Northern Tablelands ?? of ?? males, sang the Northern Tablelands song and 6 males sang it elsewhere.’

53) Reworded as suggested, replacing percentages with the actual numbers of birds singing each song type. (Line 201).

54) L.180 See general comment above on song similarity as evidence of learning. An accurate and neutral reporting of the data would be ‘Throughout the study area, 17? males sang interspecific songs: 4 resembled songs of noisy friarbird, 5 little wattlebird,’ etc etc.

54) We have fully reworded this section as suggested, and reordered the descriptions based on species frequency. The reworded results are as follows: “Located throughout the study area, eighteen males sang interspecific songs: five males’ songs resembled songs of little wattlebird *Anthochaera chrysoptera*, four of noisy friarbird *Philemon corniculatus*, two each of spiny-cheeked honeyeater *Acanthagenys rufogularis* and pied currawong *Streptera graculina*, and singles of eastern rosella *Platycercus eximius*, little friarbird *Philemon citreogularis*, olive-backed oriole *Oriolus sagittatus* and black-faced cuckooshrike *Coracina novaehollandiae* (Figures 1A & 2F-O).” (Line 210).

55) L.193 “significantly” implies a supporting statistical analysis, replace with ‘noticeably’

55) We have replaced ‘significantly’ with ‘noticeably’ as suggested. (Line 233).

56) L.197 To be clearer could replace “Repeat recordings ...” with ‘Recordings of the same individual within a year were ...’

56) We have reworded as: ‘Song recordings of the same individual over time were more similar to each other than to those of all other individuals.’ We say over time instead of within a year because some of the songs included in the Mantel test were recorded in different years (detailed in the following lines of the results section). (Line 237).

57) L.199 “remained consistent” see general comment above.

57) We have reworded this section as follows: “We recorded two colour-marked males in different years; one male produced the typical Blue Mountains song type in 2015 and 2017, and another produced the clipped Blue Mountains song type in 2016 and 2017 (Supplemental figure S3). Two males first recorded in the wild producing a typical Blue Mountains song in 2019 maintained this song type in captivity at least 18 months later, having been recruited to the captive population.” (Line 239).

58) L.203 Which males are “the other 21 individuals in the same season” referring to? Are these the captive population? This detail needs to be clarified and more obviously related to the surrounding text.

58) The ‘other 21’ refers to the other birds whose songs were included in the Mantel test (n = 25). We have clarified as follows: “We obtained repeat recordings of the songs of 21 individuals in the same season.” (Line 244).

59) L.204 “field-validated” see general comment above.

59) We have expanded to clarify as follows: “We also observed a further five colour-marked males across years, whose songs we could not record but could consistently assign in the field to the typical Blue Mountains song type.” (Line 245).

60) L.224 Can combine sentences “... with a circle. The size ... corresponds” as ‘... with a circle corresponding’.

60) Changed as requested. (Line 265).

61) L.227 Can remove “spatial windows of <”.

61) Removed as suggested. (Line 268).

62) L.216 and throughout captions, ensure label e.g., (1), (A) occurs before the item they refer to.

62) Legends to Figures 1, 2 and S5 corrected so lettering or numbering comes before the item.

63) L.232, 233 Also for subsection labels (A-E) Species-specific etc.

63) Corrected, see response to comment 62.

64) L.237 see general comment above on axis scaling for comparisons.

65) We have scaled all the spectrograms on Figure 2 to allow direct comparisons.

65) L.275 *Add information on what the line with shading indicates in (C) and (F).*

65) We have added to the figure legend “Lines and shading in C and F denote model predictions and 95% confidence intervals from the logistic regression, respectively.” (Line 323).

66) L.325 *Add of total interspecific (Five out of 18 of the interspecific singing ...) to aid interpretation.*

66) Added ‘18’ as requested. (Line 327).

67) L.343 *Is it known whether Hawaiian honeycreepers became fragmented and widespread (like regent honeyeaters) or concentrated into a smaller area of suitable habitats? If the latter, this could explain song convergence.*

67) The latter is the case for the Hawaiian honeycreepers, though we are reluctant to expand this section of the discussion with the aim of keeping the manuscript within word limits.

68) L.353-355 *Perhaps replace, or add, a comment on the practical conservation value – indications of a loss of culture may be a useful conservation tool to establish the level of threat faced by declining populations?*

68) We have added the following sentence to the end of the final paragraph: “Monitoring song cultures in wild populations may provide a useful indicator of population trajectory or threat status in species whose populations are otherwise very challenging to monitor directly.” (Line 402).

Appendix B

Australian
National
University

Fenner School of Environment and Society
Linnaeus Way
Canberra
Australia 2601
13th February 2021

Dear Dr Dall,

Many thanks indeed to yourself and reviewer two for the rapid decision to accept our manuscript titled 'Loss of vocal culture and fitness costs in a nomadic songbird' for publication.

Please find accompanying the final minor revisions to the manuscript as requested. Below in bold are our responses to each numbered comment detailing how we have dealt with each comment in the revised version of the manuscript.

We very much look forward to seeing the published version in *Proceedings B*.

Yours sincerely,

Dr Ross Crates (on behalf of the co-authors). www.difficultbirds.com

Comments from Reviewer 2:

1) l. 96 “dated” would be better.

1) Changed ‘date’ to ‘dated.’ (Line 89).

2) l. 115-118 (response to 14) and 15)) A small change to the wording at two places in this part of the text would maintain the distinction between what you observed and what you have inferred (and perhaps also fulfil the editor’s request to moderate claims).

- l.115 “produced the song of as different species” is an inference, “sang songs we considered similar to a different bird species” is what you observed.

- l.117 “the species whose songs they had learned” is an inference, “the species we considered most similar” is what you observed.

2) We have reworded as requested: ‘sang songs we considered similar to a different bird species.’ (Line 107) and ‘we considered most similar.’ (Line 110).

3) l.118 add initials in brackets to identify the experienced observer ?RC, as you have done elsewhere.

3) RC added (Line 111).

4) l.230 check dates, it says 1986-2012 at line 96. In Fig captions l.264 and elsewhere (e.g. l.275 and S2 text) it says pre-2012.

4). We have corrected the date to say 1986-2011 (Lines 89 and 339).

5) l.391 (also 67) and response to 67)) It seems a shame not to enlighten the reader on a likely reason for the difference with honeyeaters. 5 extra words could do it e.g. “ ... as the population declined, possibly because of range contraction”

5) We have added ‘...., possibly due to range contraction.’ (Line 367).

6) l.454, 531, S5 lines 1,3,7 typo: replace “signing” with “singing”.

6) Typos corrected to ‘singing’ (Line 432, 507 and in Supplemental Information).

Supplementary Information:

7) S1 para 2 line 5 change "... juvenile males there therefore isolated from ..." to "... juvenile males are therefore isolated from ...".

7) Corrected from 'there' to 'are.'

8) S1 para 2 line 10 change "...within earshot of the sounds multiple other bird species ..." to "...within earshot of the sounds of multiple other bird species ...".

8) We have added 'of' in this sentence.

9) Table S6, left hand column head change ""Xeno-canto to "xeno-canto".

9) xeno-canto decapitalized as requested.

10) Fig. S3 Present spectrograms with same x-axis scaling as Fig.2.

10). Both spectrograms now shown to x-axis scale of 2.5 seconds, same as Fig. 2.